# Armadillo-repeat kinesin1 interacts with Arabidopsis atlastin RHD3 to move ER with plus-end of microtubules

Jiaqi Sun [1,4], Mi Zhang [1,2,4], Xingyun Qi [1,3], Caitlin Doyle[1] & Huanquan Zheng [1✉]

In living cells, dynamics of the endoplasmic reticulum (ER) are driven by the cytoskeleton motor machinery as well as the action of ER-shaping proteins such as atlastin GTPases including RHD3 in Arabidopsis. It is not known if the two systems interplay, and, if so, how they do. Here we report the identification of ARK1 (Armadillo-Repeat Kinesin1) via a genetic screen for enhancers of the rhd3 mutant phenotype. In addition to defects in microtubule dynamics, ER organization is also defective in mutants lacking a functional ARK1. In growing root hair cells, ARK1 comets predominantly localize on the growing-end of microtubules and partially overlap with RHD3 in the cortex of the subapical region. ARK1 co-moves with RHD3 during tip growth of root hair cells. We show that there is a functional interdependence between ARK1 and RHD3. ARK1 physically interacts with RHD3 via its armadillo domain (ARM). In leaf epidermal cells where a polygonal ER network can be resolved, ARK1, but not ARK1ΔARM, moves together with RHD3 to pull an ER tubule toward another and stays with the newly formed 3-way junction of the ER for a while. We conclude that ARK1 acts together with RHD3 to move the ER on microtubules to generate a fine ER network.

---

[1] Department of Biology, McGill University, Montreal, Quebec H3A 1B1, Canada. [2] Biotechnology Research Center, Southwest University, Chongqing 400715, China. [3] Department of Biology, Rutgers University, Camden, NJ 08103, USA. [4] These authors contributed equally: Jiaqi Sun, Mi Zhang. ✉email: hugo.zheng@mcgill.ca

The endoplasmic reticulum (ER) is the largest organelle in eukaryotic cells. It comprises an interconnected membrane system of cisternal sheets and tubules. The ER is extensively distributed in cells from the nuclear envelope to the cell periphery. In living cells, the network of the ER is highly dynamic and undergoes continuous reorganization, including tubule formation, extension, fusion, and sliding for ring closure[1,2]. These dynamics allow the ER to perform diverse functions, including protein synthesis, folding, and sorting, as well as the production of the endomembrane system, lipid synthesis, and calcium homeostasis[2,3].

It has been long known that, in living cells, the dynamics of cytoskeletons contribute to the architecture and dynamics of the ER. In mammalian cells, microtubules (MTs) have been documented to play a key role in ER dynamics[4]. At least two types of MT-dependent ER dynamics, termed tip attachment complex (TAC) and ER sliding, have been defined[3]. In respect to TAC, ER tubules grow synchronously with MTs at their plus-end tips. Stromal interaction molecule 1 (STIM1) and end-binding1 (EB1), which localize to the tips of growing ER tubules and the MT plus end, respectively, form a complex to lead this process[5]. To date, the motor for this movement is not defined. In respect to the ER-sliding mechanism, the movement of ER tubules can be driven by MT motors (kinesin 1 and dynein) along acetylated MTs[6,7]. In contrast to what is described in mammalian cells, it has been evidenced that in plant cells, ER dynamics is largely determined by the actin–myosin machinery[8–11]. For example, a defect of myosin XIs, especially myosin XI-K and XI-2, dramatically suppresses the dynamics of the ER, causing an aberrant ER organization[11]. Currently, limited data suggested that microtubules also play a role in the formation of the ER network, in particular, in elongating cells[12] and rapid tip growing root-hair cells[13]. In hypocotyl epidermal cells, microtubules contribute to ER tubule elongation at a slower rate than the actin–myosin system[14]. However, no MT–ER connecting molecules have been identified in plant cells.

In addition to the role of the cytoskeleton, action of some ER-shaping proteins has recently also been illustrated in ER dynamics. For example, a family of reticulon proteins has been shown to be involved in the formation of ER tubules[15]. These reticulon proteins have a W-shaped topology and are inserted into the cytoplasmic leaflet of the ER membrane bilayer with their four transmembrane domains to generate the curvature required for the tubule formation[16–18]. Atlastin-related GTPases including ROOT HAIR DEFECTIVE3 (RHD3) in Arabidopsis[19] play a key role in mediating the homotypic fusion of ER tubules to generate three-way junctions of the ER[20,21]. Atlastin GTPases anchored in opposing membranes dimerize to tether two different membranes and then undergo a conformational rearrangement that fuses the two membranes together[22,23]. Although it is reported in mammalian cells that, atlastin interacts with spastin[24], a microtubule-serving protein[25], the exact way how atlastins as well as other ER-shaping proteins interplay with the cytoskeleton during the formation of the interconnected ER network has not been well studied.

Arabidopsis rhd3 mutant plants grow short and wavy root hairs[26,27] because of a defect in targeted secretion of secretory vesicles, resulted from the disorganized ER[13]. Because the root-hair growth defect can be easily detected and quantified[13], the rhd3 mutant becomes an ideal genetic model for further investigating how this ER fusogen may work together with other factors in the generation of the ER network during polarized cell growth. We report here the isolation and characterization of two alleles of an rhd3-1 enhancer named rhd3 enhancer9 (ren9). Root hairs of rhd3-1 ren9-1 double-mutant plants are curled and shorter than those of rhd3-1. REN9 encodes an armadillo-repeat-

containing kinesin called ARK1, which promotes microtubule catastrophe[28]. In addition to the reported defect in the organization of microtubules[29,30], ren9 also has defects in the organization of the ER. By in vivo imaging, chemical and protein–protein interaction analyses, we show that ARK1 interacts with RHD3 via its armadillo domain to lead the growth of the ER tubule together with the growing tip of microtubules for the generation of a fine ER network in plant cells.

## Results

### Isolation of rhd3 enhancer9 (ren9).
Arabidopsis rhd3 mutant plants produce short and wavy root hairs[26,27] (Fig. 1c, g). To identify factors that work together with RHD3, we carried out an

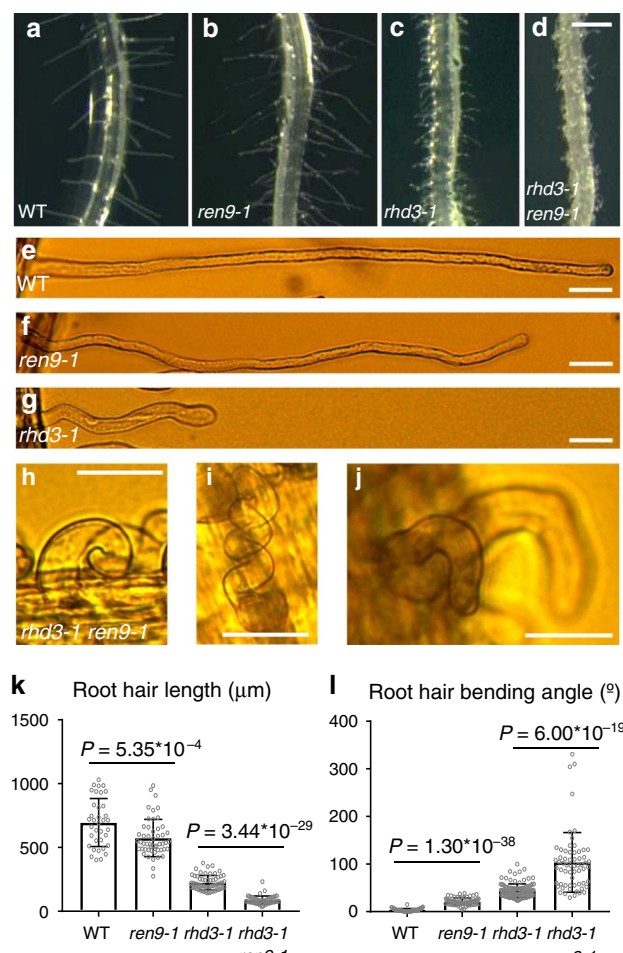

**Fig. 1 Isolation of ren9 as an rhd3 enhancer. a–d** Overview of root-hair phenotypes. Root segments of 5-day-old seedlings are shown. Scale bars = 0.5 mm. **e–j** Enlargements of representative root hairs. **a, e** wild-type; **b, f** ren9-1; **c, g** rhd3-1, **d, h–j** rhd3-1 ren9-1. Note that root hairs of rhd3-1 and ren9-1 were both wavy, but root hairs of rhd3-1 rend9-1 were curly or helical. Scale bars indicate 50 μm. **k–l** Quantification of the length (**k**) and bending angle (**l**) of root hairs. Note that the root-hair length of ren9-1 was about 80% of wild-type, while the root-hair length of rhd3-1 ren9-1 was <40% of rhd3-1 (**k**). The bending of ren9-1 related to wild-type was ~18°, while the bending of rhd3-1 ren9-1 related to rhd3-1 was about 60° (**l**). Columns = mean values, error bars = standard errors. Gray circles represent actual data. P values indicated for significances between samples are calculated by Student's t test (two-sided). In the order of wild-type, ren9-1, rhd3-1, and rhd3-1 ren9-1, n (individual root hairs) = 36, 51, 60, 52 for root-hair length, n (individual root hairs) = 61, 75, 118, and 62 for root-hair bending angle, respectively.

EMS-based screen for genetic modifiers of *rhd3-1*. Two alleles of a monogenic and recessive enhancer, denoted as *rhd3 enahncer9 (ren9)*, were identified. Root hairs in *rhd3-1 ren9-1* were curly or helical (Fig. 1d, h–j). Consequently, root hairs of *rhd3-1 ren9-1*, unlike *rhd3-1* (Fig. 1c, g) and wild-type (Fig. 1a, e), were huddled on the surface of roots (Fig. 1d). Root hairs of segregated *ren9-1* were wavy and slightly shorter than that of wild-type (Fig. 1b, f). The root-hair phenotype of *ren9-2* was identical to *ren9-1*, and the cross between the two alleles did not complement each other (Supplementary Fig. 1). Our quantification of the root-hair length and bending angle of *ren9-1* vs. wild-type and *rhd3-1 ren9-1* vs. *rhd3-1* (Fig. 1k–l) indicated that *ren9* synergistically enhanced the *rhd3* defect in the extent and direction of root-hair cell growth.

The direction of root-hair growth can be monitored by YFP-RAB-A2a, a secretory vesicle marker[31] and YFP-CLSD3, a glycan synthase that localizes to the apical plasma membrane of growing root hairs[13,32]. In wild-type, the YFP-RAB-A2a signal was enriched at the tip of growing root hairs with a rapid swing from side to side in the apical dome, so the directional tip growth was maintained[13] (Fig. 2a, e). In contrast, in *rhd3-1*, the rapid swing of YFP-RAB-A2a signal was suspended sometimes when root hairs were in bending growth (17.5′ to 24.0′, Fig. 2b, e). During this time, the maximum point of YFP-RAB-A2a stayed at the side corresponding to the growth direction (Fig. 2b). Similar suspensions were observed in root hairs of *ren9-1* (7.5′ to 9.0′ and 13.5′ to 15.5′, Fig. 2c, e). In *rhd3-1 ren9-1*, the rapid swing of YFP-RAB-A2a signal was totally abolished, and the maximum point always maintained at the side of the direction of root-hair growth (Fig. 2d, e). Altered positioning of secretion marked by RAB-A4b was also described in *rhd4*[33], and was proposed to be due to altered positioning of the tip-localized expansion zone[34]. Similar oscillations for YFP-CSLD3 in wild-type, *rhd3-1*, *ren9-1*, and *rhd3-1 ren9* were also observed (Supplementary Fig. 2). These results confirm that *ren9* enhanced the defect of *rhd3* in the direction of root-hair cell growth.

**REN9 encodes armadillo-repeat kinesin 1 (ARK1).** To identify the gene responsible for the described phenotypes in the *rhd3 ren9* double mutant, we carried out a mapping-based cloning in F₂ seedlings of *ren9-1 × Ler* and roughly mapped the *ren9* locus to the chromosome 3 of Arabidopsis. In this chromosome, *ARK1* is a characterized gene, and *ark1* mutant plants exhibit root-hair phenotypes like *ren9*[30,35]. Thus, we sequenced the genomic DNA of *ARK1* in *ren9-1* and *ren9-2*, which found a C-to-T mutation at nucleotide 334 in *ren9-1* (in exon 2) and another C-to-T mutation at nucleotide 858 in *ren9-2* (in exon 5) (Fig. 3a). Both transitions resulted in premature termination of the translation in the motor domain (Fig. 3a). To further confirm *ren9-1* is an allele of *ark1*, two T-DNA insertion mutant *ark1-1* (Salk_035063c) and *ark1-2* (Salk_081412) were crossed with *ren9-1*, respectively. All resulting F₁ seedlings still showed *ark1* wavy root hairs (Supplementary Fig. 3a–f). Furthermore, when *rhd3-1* was crossed with *ark1-2* (Supplementary Fig. 32h), *rhd3-8* (Salk_025215) was crossed with *ren9-1* (Supplementary Fig. 3i) or *ark1-2* (Supplementary Fig. 3j), all newly generated double mutants showed curly or helical root hairs on the root surface like *rhd3-1 ren9-1* (Supplementary Fig. 3g). Taken all together, *ren9* is an allele of *ark1*.

Next, the genomic DNA of *ARK1* including its promotor was cloned, tagging with *GFP* as described[28] and transformed into the *ren9-1* mutant to generate stable transgenic plants. The expression of ARK1pro:gARK1–GFP (hereafter it is referred to as gARK1–GFP) in *ren9-1* fully rescued root-hair phenotypes of *ren9-1* (Fig. 3b, d). This molecular complementation further confirms that *REN9* encodes ARK1 and that this C-terminal GFP tagged ARK1 is fully functional in vivo.

**ARK1 is involved in the formation of a fine ER network.** ARK1 is a microtubule plus-end kinesin protein that is involved in microtubule catastrophe in the organization of fine microtubules[28]. The synergistic genetic interaction between *ARK1* and *RHD3* promoted us to investigate if ARK1 plays a role in the formation of a fine ER. To this end, we examined the ER morphology in the *ren9-1* and *rhd3-1 ren9-1* double mutants using YFP-Sey1p⁶⁸²⁻⁷⁶⁶, which is the transmembrane domain and C-terminal extension of Sey1p (a yeast homolog of RHD3) and has been used as an ER marker in plant cells[36]. We did not use the commonly used ER marker GFP-HDEL, as its expression produces ER aggregates in growing root hairs[37]. YFP-Sey1p⁶⁸²⁻⁷⁶⁶ marked a subapex-focused gradient of fine ER in growing root hairs of wild-type (Fig. 4a) as reported[13]; such distribution was maintained as the root hair elongated (Fig. 4b). Thick ER bundles were hardly visible, as fluorescence signal was evenly distributed in the shank region of the root hair (Fig. 4c). In *rhd3-1*, thickly bundled ER tubules close to the concave side of the root hair were visible[13] (Fig. 4d, arrowhead). As the root hair bent to another side, the extended ER bundle shifted to the next concave side[13] (Fig. 4e, arrowhead). This bundle shift was evident with the quantified fluorescence (Fig. 4f). Interestingly, the ER morphology in *ren9-1* was also affected, although less prominent than that in *rhd3-1*. Close to the concave side, loose bundles of the ER were observed (Fig. 4g, arrowhead). The newly formed ER bundle also shifted to another side along with the growth of the root hair (Fig. 4h, arrowheads), supported by the quantified fluorescence (Fig. 4i). In *rhd3-1 ren9-1*, the ER defect was more severe than those in either single mutant. The ER dramatically aggregated into a mass from which some ER bundles extended outward (Fig. 4j, arrowhead). The ER mass was firmly close to the concave side of the root hair and never extended to another side (Fig. 4k, arrowhead) during root-hair growth, as the quantification indicated (Fig. 4l). These results suggest that ARK1 is involved in a fine ER formation in growing root hairs.

As expected, in *ren9-1*, the Z-projection of microtubules marked by the MT marker protein mCherry-MAP4[38] showed that MTs were disordered and thick bundles appeared in the endoplasm close to the concave side of root hairs (Fig. 4s, t, asterisks). In subapical flanks of the root hair, cortical MTs (cMTs) are also bundled preferentially at the cortex of convex sides (Fig. 4s, t, arrows). As root hairs bent to another direction, MT bundles shifted their localization (Fig. 4s, t, arrows; Fig. 4u). In wild-type, most MTs were cortical and arranged along the longitudinal axis of cells, but absent in the apical dome (Fig. 4m). This arrangement was maintained as the root hair elongated (Fig. 4n). The quantification of fluorescence indicated that there was no thick bundles and a shift in wild-type (Fig. 4o). Interestingly, in *rhd3-1*, MTs were arranged like that observed in *ren9-1* (Fig. 4p, q), MTs were bundled in the endoplasm (Fig. 4p, q, asterisks), and cMT were also bundled at the cortex of convex sides (Fig. 4p, q, arrows). There was also a shift of MT bundles (Fig. 4r). In *rhd3-1 ren9-1*, MT bundles were much more apparent, the localization of MT bundles (both in the endoplasm and cortex) did not shift to another side of the root hair (Fig. 4v, w, asterisks and arrows; Fig. 4x) during the growth of root hairs. These results suggest that RHD3 is also involved in the microtubule organization.

**ARK1 overlaps and co-moves with cortical RHD3.** Next, we would like to understand how ARK1 could be involved in the formation of a fine ER network. ARK1 is known to be localized to

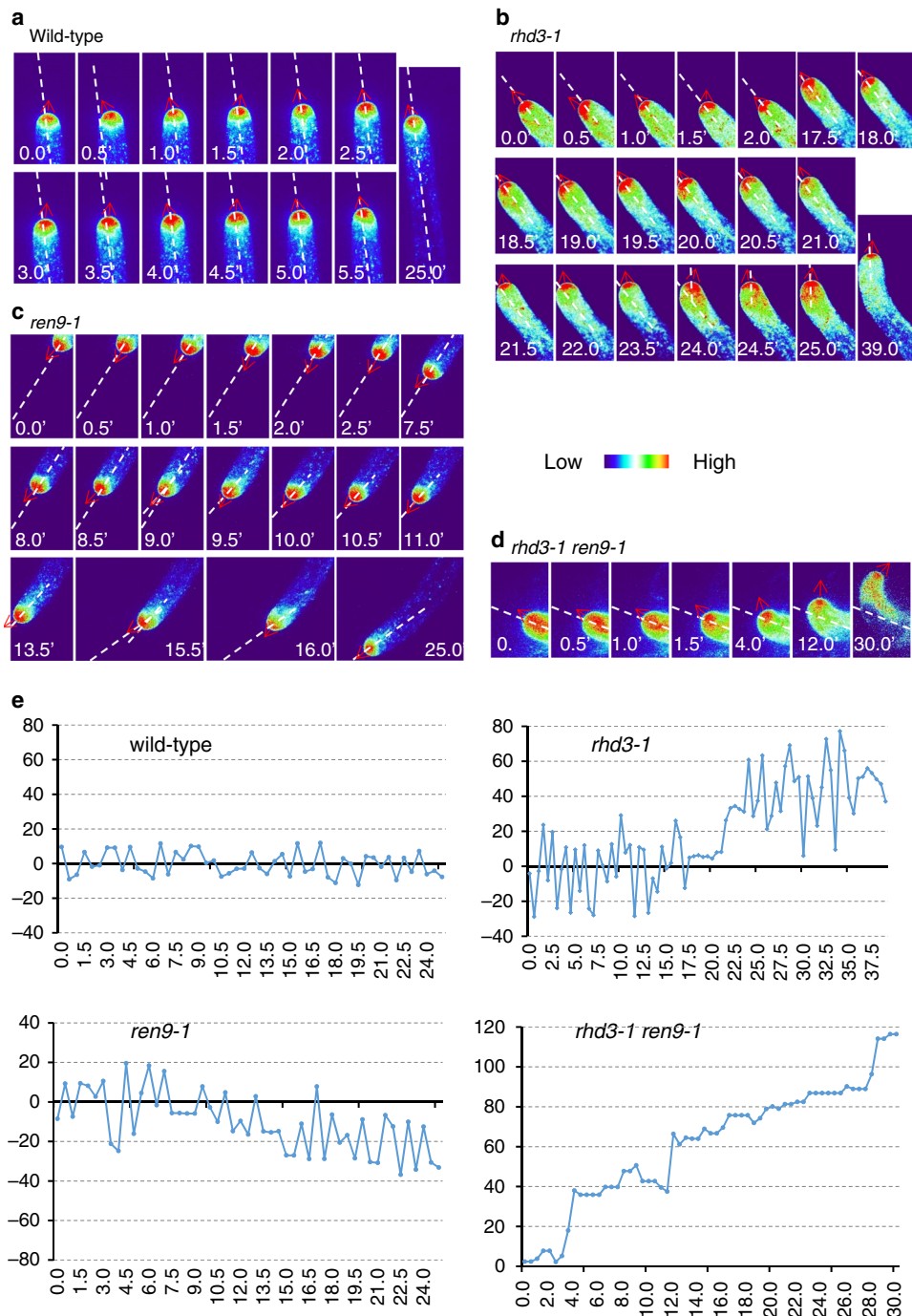

**Fig. 2 Targeting of YFP-RAB-A2a at the tips of growing root hairs. a–d** Heatmap of the targeting of YFP-RAB-A2a in a root hair of wild-type (**a**), *rhd3-1* (**b**), *ren9-1* (**c**), and *rhd3-1 ren9-1* (**d**). **e** Quantification of swings of YFP-RAB-A2a targeting related to the previous growth direction (marked by white dashed lines in **a–d**). YFP-RAB-A2a signals are shown in the heatmap format. The color code illustrates the signal intensity. Red arrows indicate the targeting direction of YFP-RAB-A2a at the tips. White dashed lines present the previous direction of root-hair growth. The experiment was independently repeated six times. In wild-type, there were rapid swings of YFP-RAB-A2a (**a**, **e**); in *rhd3-1*, rapid swings were suspected from 18' to 39' (**b**, **e**) at which the root hair started to bend. In *ren9-1*, there were also suspensions of rapid swings at 7.5'-9'and 11'-15.5' (**c**, **e**). In *rhd3-1 ren9-1*, there was no swing at all; YFP-RAB-A2a was targeted toward the bending direction (**d**, **e**). The scale bars = 10 μm.

the plus end of growing MTs[28]. We found that gARK1–GFP was preferentially expressed in growing root hairs, and was mainly distributed as comets or punctae in the cortex of the subapical region of growing root hairs (Fig. 5a, arrows and asterisks), but it is missing in the apical dome (Fig. 5a). We noted that this localization pattern is different from the previous finding[28,39]. The discrepancy may arise from different growth ages of root-

hair cells examined and/or different expression levels between transgenic lines used. When co-expressed with mCherry-MAP4, gARK1–GFP was found to be localized largely at the growing plus end of MTs in the cortex of the subapical region (Fig. 5b, arrowheads and Supplementary Movie 1). Our 3D modeling (Fig. 5c and Supplementary Movies 2–4) from a z-stack further confirmed this notion. It has been recently reported[13] that, in

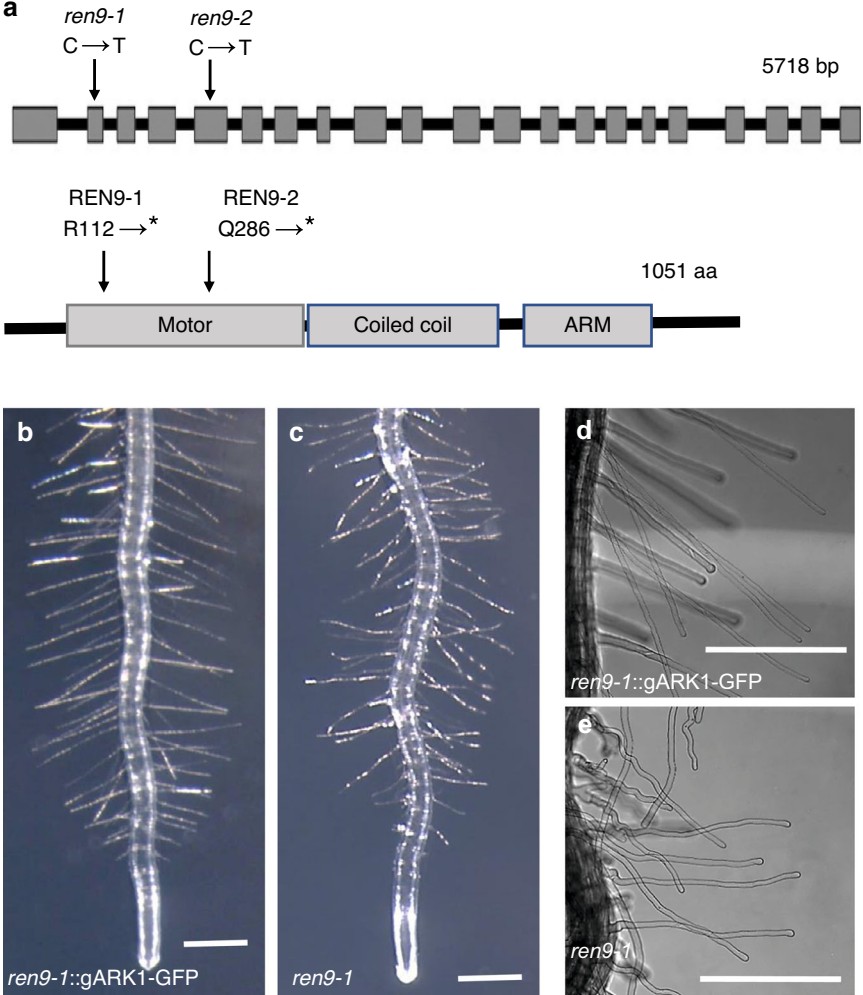

**Fig. 3 Cloning of the REN9/ARK1 gene.** Cloning of the REN9/ARK1 gene. **a** Schematic representation of the REN9/ARK1 genomic structure and protein domain organization. The genomic structure of 5718 bp is shown with exons (gray rectangles) and introns (black lines in the gene diagram). The ARK1 protein of 1051 aa is shown with motor, coiled coil and armadillo (ARM) domains (gray rectangles). Arrows indicate the single-base transition and resulting change of the amino acid residue in ren9-1 and ren9-2. Asterisks indicate the stop codon. (**b**–**e**) gARK1–GFP rescues the root-hair defect of ren9-1. Root hairs of transgenic (**b**, **d**) and non-transgenic ren9-1 (**c**, **e**). Scale bars = 0.5 mm.

growing root hairs of wild-type plants, there was a subapex-focused gradient of RHD3, which is also missing in the apical dome (Fig. 6a, b). Therefore, we asked how the distribution of cortical ARK1 may relate to subapex-focused RHD3 during tip growth of root-hair cells. When gARK1–GFP was co-expressed with RFP-RHD3, we found that the signal of cortical gARK1-RFP surrounded the subapex-focused RFP-RHD3 (Fig. 6a, b) and co-moved with it in the cortex of the subapical region as the root hair elongated (Supplementary Movies 5 and 6). The multichannel kymograph analysis confirmed this co-movement relationship as root-hair elongated: when the gARK1–GFP signal was dim (Fig. 6c, arrow), the RFP-RHD3 signal was also weak (Fig. 6c, arrows). We noted there were overlapped signals between gARK1–GFP and RFP-RHD3 in the cortex of the subapical region as root-hair elongated (white signals in merged image in Fig. 6b and in Supplementary Movie 6). To better illustrate the spatial relationship between ARK1 and RHD3 in the cortex of the subapical region, we did a 3D modeling from a z-stack of the growing root hair showing in Fig. 6b. We found that a part of cortical ARK1 clearly overlapped with cortical RHD3 in the cortex of the subapical region (Fig. 6d and Supplementary Movies 7–9), suggesting that ARK1 may lead to the growth of ER tubules in growing root hairs.

**ARK1 and RHD3 are functionally interdependent.** We then wondered what the causal relationship is between ARK1 and RHD3. As described[13], in growing root hairs of wild-type, YFP-RHD3 was focused in the subapical region and missing in the apical dome (Fig. 7a and Supplementary Movie 10). We found that, in growing root hairs of ren9-1, YFP-RHD3 was no longer restricted in the subapical region (Fig. 7a and Supplementary Movie 11). The quantification of the height of the apical dome and the thickness of the subapical YFP-RHD3 signal indicated that YFP-RHD3 moved into the apical dome and lagged behind the subapical region in ren9-1 (Fig. 7b, c). There were also YFP-RHD3 bundles visible in the endoplasm (Fig. 7a, arrowheads). In wild-type, the maximal and average velocity of RHD3 streaming were ~0.9 and ~0.3 μm/s, respectively, while these were reduced to ~0.7 and ~0.2 μm/s in ren9-1 (Fig. 7d). In addition, in growing root hairs of wild-type treated with taxol (1 μM), an MT-stabilizing drug that antagonizes the action of ARK1[28] produced a phenotype similar to what was observed in ren9-1 that, the subapex-focused distribution of YFP-RHD3 was also disrupted (Supplementary Fig. 4a). Bundles of YFP-RHD3 were also visible in the shank region of the root hair (Supplementary Fig. 4a, arrowheads). Both the maximal ($P = 2.04 \times 10^{-5}$) and average ($P = 1.59 \times 10^{-15}$) velocities of RHD3 streaming were significantly

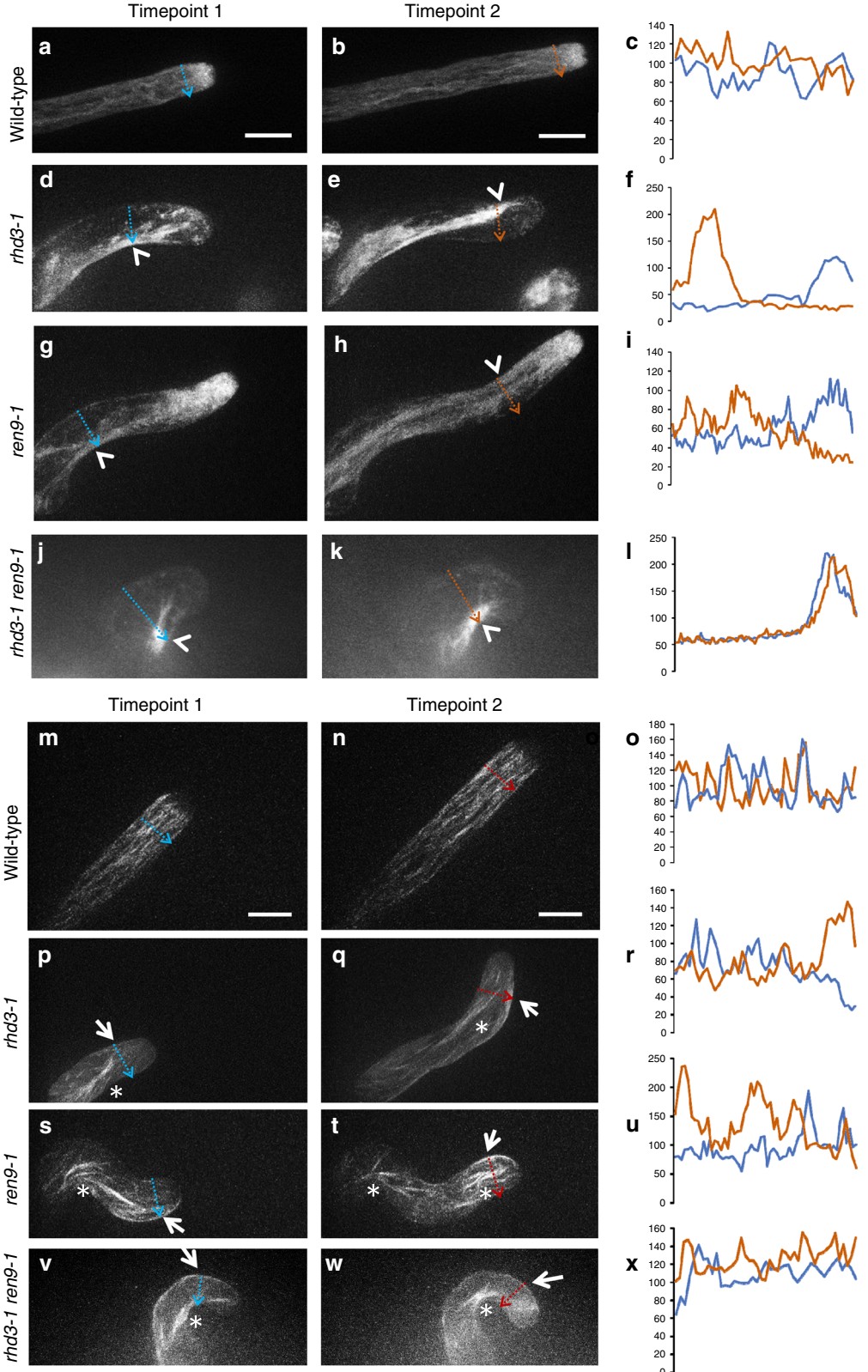

reduced in root-hair cells treated with 1 µM taxol (Supplementary Fig. 4b). In the endoplasm of non-taxol-treated root hairs, the distribution of both RHD3 and ARK1 was fine and even; bundles were hardly but occasionally observable (Supplementary Fig. 4c, arrowhead). In the endoplasm of taxol-treated root hairs, the association frequency between ARK1 and RHD3 bundles was increased (Supplementary Fig. 4c, arrowheads).

On the other hand, gARK1–GFP comets in *rhd3* root hairs exhibited a great tendency to gather at the apex (11 out of 18 root hairs) or one side of the apical shoulder (7 out of

**Fig. 4 Organization of the ER and MTs in growing root hairs. a–l** ER morphology labeled by YFP-Sey1p[682–766] during tip growth of a root hair of wild-type (**a**, **b**), *rhd3-1* (**d**, **e**), *ren9-1* (**g**, **h**), and *rhd3-1 ren9-1* (**j-k**). Arrowheads indicate ER bundles. Z-stack images were captured by spinning disc confocal microscopy with a step size of 1 μm. Z-projection are shown at two timepoints of growing root hair. **c**, **f**, **i** and **l** are fluorescence intensities of YFP-Sey1p[682–766] in blue and red lines drew in respected images. **m–x** The microtubule organization labeled by mCherry-MAP4 during tip growth of a root hair of wild-type (**m**, **n**), *rhd3-1* (**p**, **q**), *ren9-1* (**s**, **t**), and *rhd3-1 ren9-1* (**v**, **w**). Arrows indicate cortical MT bundles, and asterisks indicate endoplasmic MT bundles. Scale bars indicate 10 μm. Z-stack images were captured by spinning disc confocal microscopy with a step size of 0.5 μm. Z-projection images are shown at two timepoints of growing root hair. **o**, **r**, **u**, and **x** are fluorescence intensities of mCherry-MAP4 in blue and red lines drawn in respected images. Both experiments were independently repeated six times. Source Data are available.

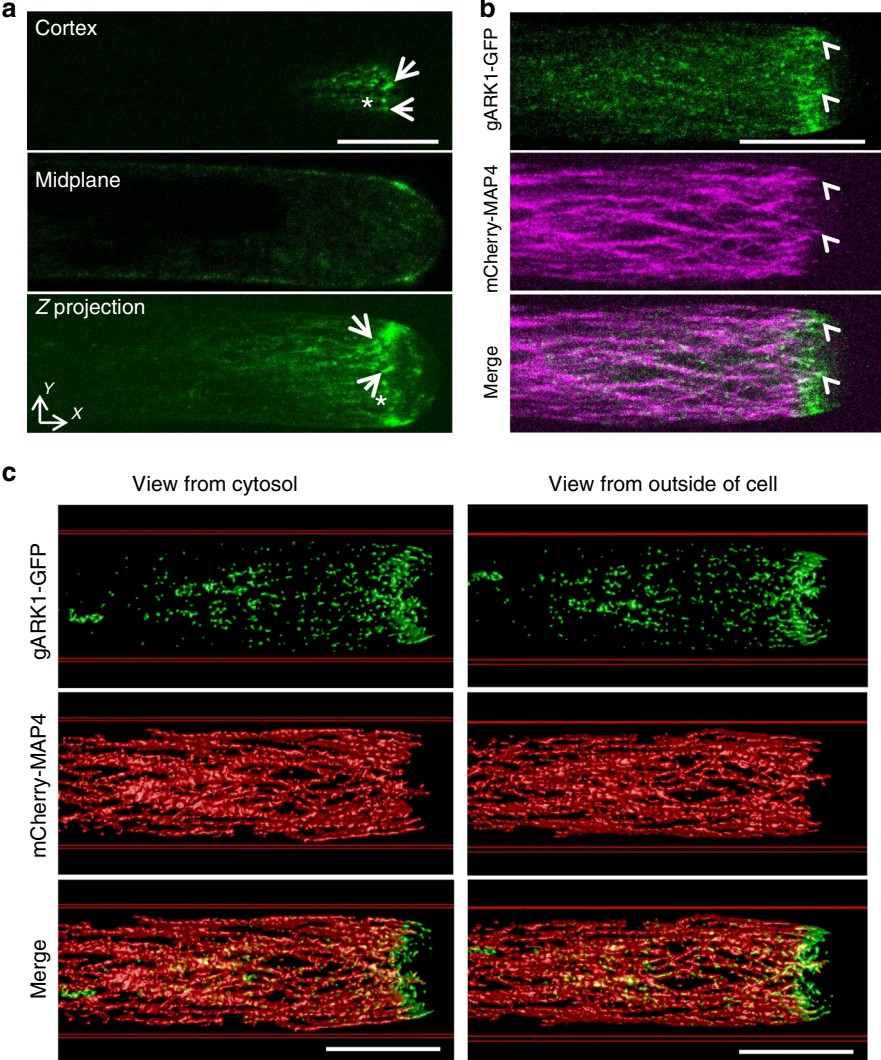

**Fig. 5 ARK1 localizes to growing ends of microtubules in the cortex of the subapical region of growing root hairs. a** Localization of gARK1–GFP in growing root hairs. Distribution of gARK1–GFP in the cortex, midplane, and Z-projection of a growing root hair. Arrows indicate gARK1–GFP comets; asterisks indicate gARK1–GFP punctae. gARK1–GFP was introduced into *ren9-1* and imaged in growing root hairs of 5-d seedlings. **b** Distribution of gARK1–GFP and MTs labeled by mCherry-MAP4 in a growing root hair. Arrowheads indicate the MT plus-end localization of gARK1–GFP. Both experiments were independently repeated six times. **c** Spatial relation of gARK1-RFP (top and merged panels) and mCherry-MAP4 (middle and merged panels) in a 3D model. The left image is a view from the cytosol; the right is a view from the outside of the cell. Scale bars indicate 10 μm.

18 root hairs) (Fig. 7e, lower panel), comparing to subapex-focused gARK1–GFP in *ren9-1* (Fig. 7e, upper panel). Given the slightly swollen morphology observed when gARK1–GFP was localized at the extreme apex (Fig. 7e), it seemed that the root hair ceased its tip restricted elongation at the moment. Furthermore, the movement of gAKR1-GFP in *rhd3* was randomized (Fig. 7f, g and Supplementary Movies 12, 13), although the velocity of

gARK1–GFP in *rhd3-1* was not changed (Fig. 7h). Taken all together, we think that ARK1 and RHD3 are functionally interdependent during the root-hair growth.

**ARK1 physically interacts with RHD3 through its ARM domain.** We then examined if ARK1 and RHD3 physically interacted with each other. Because ARK1–GFP from the native

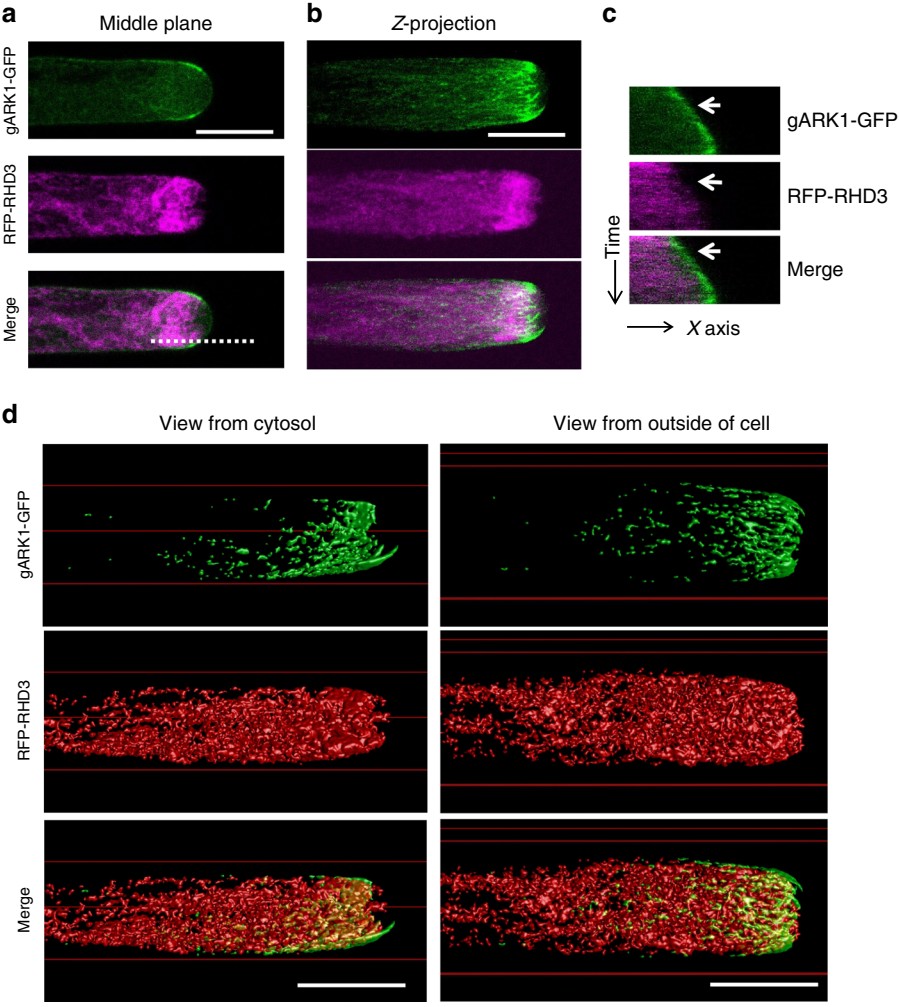

**Fig. 6 Cortical ARK1 overlaps and co-moves with RHD3 in the cortex of the subapical region of growing root-hair cells. a, b** Distribution of cortical gARK1–GFP and subapex-focused RFP-RHD3 in the midplane (**a**) and Z-projection (**b**) in the subapical region of a growing root-hair cell. The experiment was independently repeated three times. **c** Kymograph of gARK1–GFP and RFP-RHD3 in the midplane. The dash line in (**a**) show the region used for the analysis. **d** Spatial relation of gARK1-RFP (top and merged panels) and RFP-RHD3 (middle and merged panels) in a 3D model. The left image is a view from the cytosol; the right is a view from the outside of the cell. Partial overlap was visible in the cortex of the subapical region. Scale bars indicate 10 μm.

promoter was preferentially expressed in growing root hairs, it was difficult to extract enough ARK1–GFP from transgenic Arabidopsis plants expressing gARK1–GFP. The stable expression of ARK1–GFP from a ubiquitous promoter in plants is also impossible as it is lethal to plant growth[28]. So we transiently expressed mCherry-RHD3 with ARK1–GFP, ARK1ΔARM-GFP, a mutant version of ARK1 that can still mark plus end of microtubules and rescue dynamics of microtubules[39] or GFP in tobacco leaf cells at OD600 = 0.01. When ARK1–GFP was purified by using GFP-trap beads, mCherry-RHD3 was co-purified (Fig. 8a, lane 3). The ARM domain of ARK1 is considered to be a cargo domain[28]. We found that ARK1ΔARM-GFP was neither able to co-purify mCherry-RHD3 (Fig. 8a, lane 2) nor was free GFP (Fig. 8a, lane 1). We then used a 3-in-1 BiFC system where mCherry-HDEL is co-expressed[40] with the two testing genes to confirm the interaction. RHD3 interacted with full-length ARK1 at either three-way junctions, or two-way junctions, or at tips of ER tubules (Fig. 8b), as well as with the ARM domain of ARK1 (Fig. 8b), but not with ARK1ΔARM (Fig. 8b). Together, we concluded that ARK1 physically interacts with RHD3 through its ARM domain. We noted that the RHD3-ARK1 BiFC signal associated but not colocalized with the ER, whereas the RHD3-ARM BiFC signal colocalized with the ER (Fig. 8b). It may be

because with the motor domain, full-length ARK1 is stably localized on the plus end of MTs[28], so the BiFC signal labels ER-MT junctions. Without the motor domain, the association of ARM with MTs may be weakened, so it can be brought to the ER by RHD3.

**ARK1 moves with RHD3 to pull an ER tubule toward another.** Although ARK1 and RHD3 overlap and co-move in the subapical region during root-hair tip growth, detailed characterization of how ARK1 and RHD3 co-move in root hairs was difficult due to the resolution limit at the tip of growing root-hair cells and the dense cytoplasmic fluorescence of both gARK1–GFP and RFP-RHD3 (Fig. 6a, c). However, it is known that, in leaf epidermal cells, RHD3 marks a polygonal mesh network of the ER with enrichment in punctae[19], and ARK1–GFP comets and/or punctae are also observable in leaf cells when transiently expressed[28]. Therefore, we examined if ARK1 moves together with RHD3. If so, how they may move in leaf epidermal cells. To this end, ARK1–GFP and mCherry-RHD3 were transiently expressed in tobacco leaf cells at OD600 = 0.01. When transiently expressed, ARK1–GFP was able to label MTs, in addition to MT plus-end labeling[28]. Therefore, we first quantified the ratio of MT-positive and MT-negative (presumably actin–myosin dependent) ER

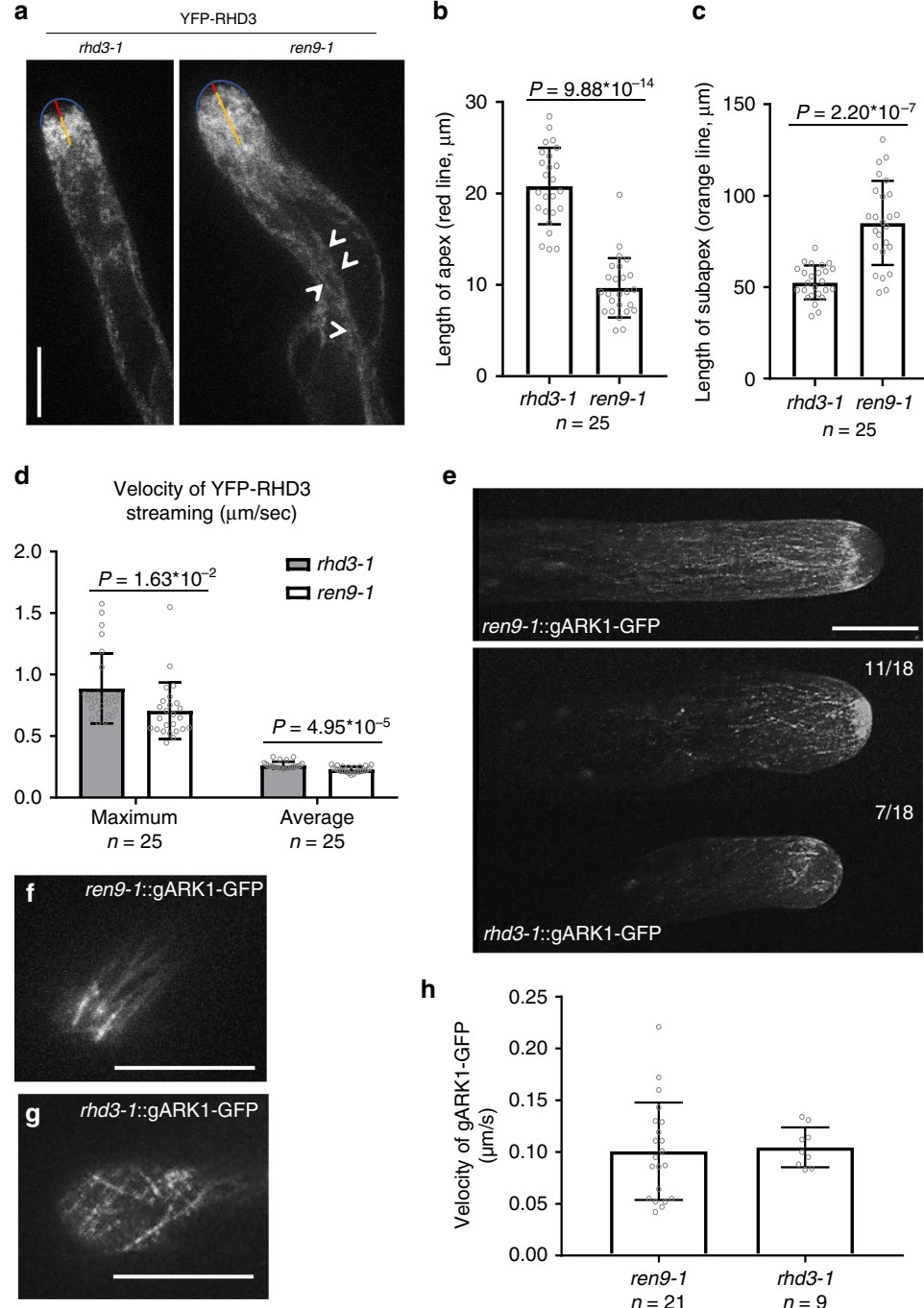

**Fig. 7 ARK1 and RHD3 are functionally interdependent. a** Distribution of YFP-RHD3 in *rhd3-1* and *ren9-1*. **b**, **c** Quantification of the length of the apical dome (**b**) and the subapical region (**c**) of *rhd3-1* and *ren9-1* expressing YFP-RHD3. Columns = mean values, error bars = standard deviations. Gray circles represent actual data. *P* values indicated for significances between samples are calculated by Student's *t* test (two-sided). *n* = individual root hairs. **d** Maximal and average velocity of RHD3 streaming in growing root hairs of *rhd3-1* and *ren9-1* expressing YFP-RHD3. Columns = mean values, error bars = standard deviations. Gray circles represent actual data. *P* values indicated for significances between samples are calculated by Student's *t* test (two-sided). *n* = individual root hairs. **e** Distribution of gARK1–GFP in root hairs of rescued *ren9-1* (upper panel) and *rhd3-1* (lower panel). Z-projection images are shown. The ratio of the representative gARK1–GFP distribution in *rhd3-1* is indicated at the upper right corners. **f**, **g** Time-projection of gARK1–GFP punctae in the cortex of a growing root hair of *ren9-1* (**f**) and *rhd3-1* (**g**). A total of 21 images were captured with an interval of 2 s. Scale bars = 10 μm. **h** Velocity of gARK1–GFP punctae in growing root hairs of *ren9-1* and *rhd3-1*. gARK1–GFP was imaged every 2 s. Columns = mean values, error bars = standard deviations. Gray circles represent actual data. *n* = individual gARK1–GFP punctae/comets. Source Data are available.

extension events in tobacco leaf epidermal cells. We found that 88/91 ER extension events (counted from four cells in two experimental replications) were MT-negative, only 3/91 were MT-positive. This indicated that, in tobacco leaf epidermal cells, MT-negative ER extension plays a major role in ER dynamics, but

MT-positive ER extension also contributes to the complexity of the ER. Hamada et al. reported using GFP-tubulin that, ER tubules elongate along preexisting microtubules toward both plus and minus ends in Arabidopsis hypocotyl epidermal cells[14]. We then focus on MT-dependent ER extension in our system. Of 116

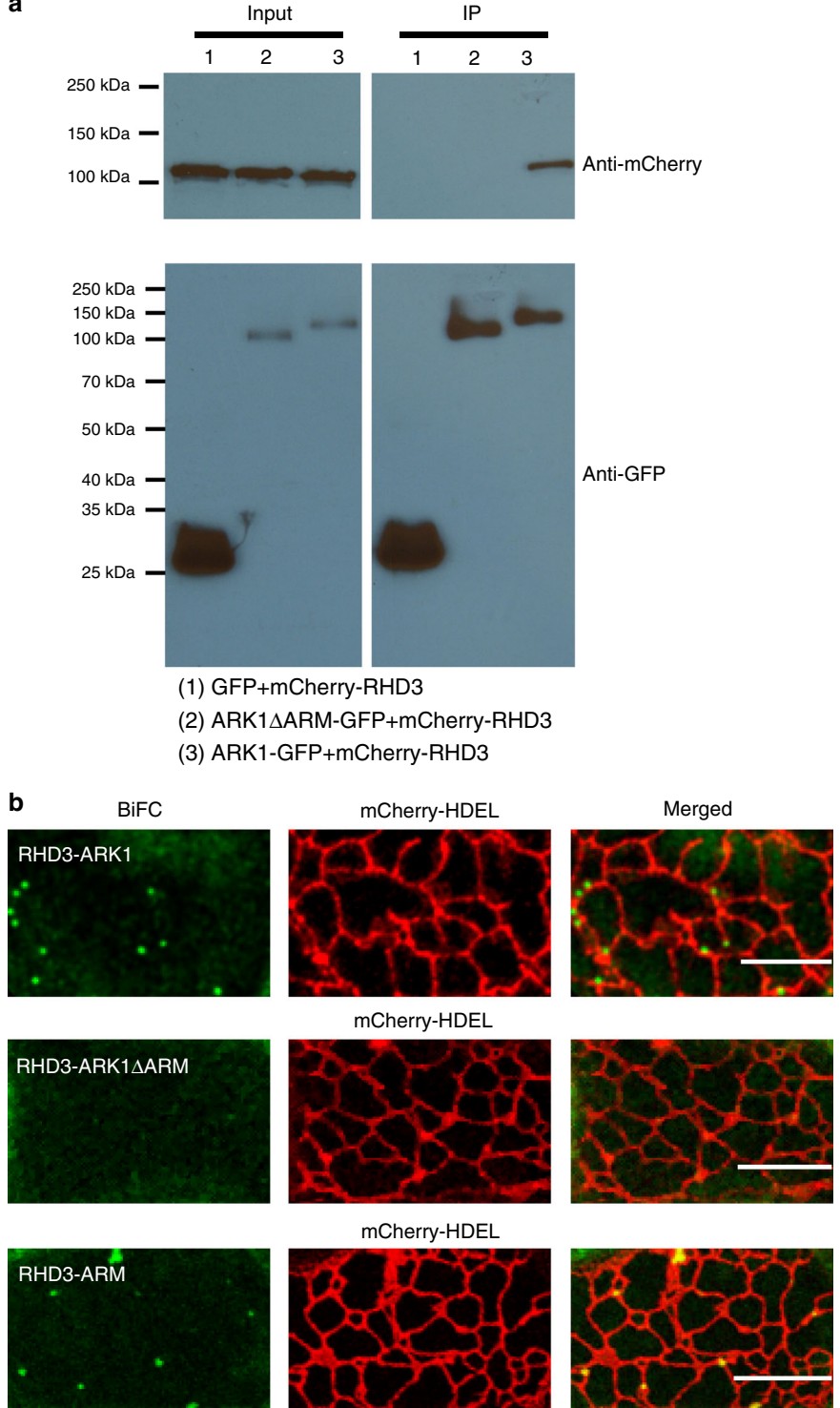

(1) GFP+mCherry-RHD3
(2) ARK1ΔARM-GFP+mCherry-RHD3
(3) ARK1-GFP+mCherry-RHD3

**Fig. 8 ARK1, but not ARK1ΔARM interacts with RHD3. a** Co-purification of mCherry-RHD3 with ARK1–GFP (line 3) but not with ARK1ΔARM (line 2) or GFP (line 1) using GFP-trap. mCherry-RHD3 was detected by anti-RFP; ARK1–GFP, ARK1ΔARM, or GFP was detected by anti-GFP. Input is a total protein extract samples; IP is purified samples with GFP-trap. **b** Venus-based Bi-molecular fluorescence complementation (BiFC) analyses of nVenus-RHD3 and ARK1-cVenus or ARK1ΔARM-cVenus or ARM-cVenus. mCherry-HDEL from the same gene cassette was used as an expression indicator. This experiment was independently repeated three times. Scale bars = 10 μm. Source Data are available.

MT-dependent ER extension events observed from 45 cells picked from 9 experimental replications, 88 were preexisting microtubule dependent, and 28 were ARK1-marked growing-end related. At the same time, we also recorded 80 ARK1 comets/punctate movements and noted that there were certain numbers of ARK1 comets/punctate (52/80) which did not move with RHD3. This may be due to the weak activity of ARK1 in leaf epidermal cells, or that other factors may compete with ARK1 to guide the action of RHD3 in leaf epidermal cells. We then focused on the ARK1-positive growing-end-dependent ER extension. We found that

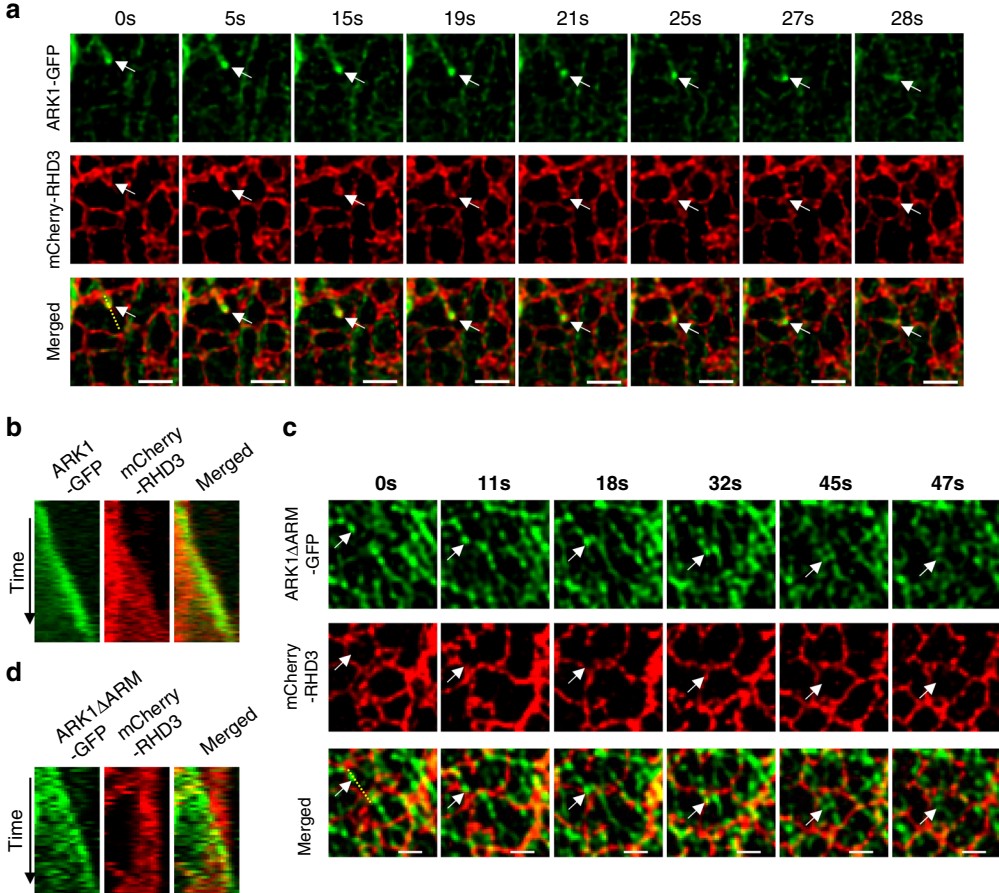

**Fig. 9 ARK1–GFP, but not ARK1ΔARM-GFP, co-moves with mCherry-RHD3 to pull an ER tubule toward another in an epidermal leaf cell. a** The co-movement of an ARK1–GFP comet/punctate (white arrows) together with mCherry-RHD3 along with the growth of an ER tubule during a time. The experiment was independently repeated nine times. Scale bars = 2 μm. **b** A kymograph of the yellow line in (**a**) showing ARK1–GFP is leading the growth of ER tubule. **c** The movement of an ARK1ΔARM-GFP comet (white arrows) in relation with mCherry-RHD3 during a time. This experiment was independently repeated four times. Scale bars = 2 μm. **d** A kymograph of the yellow line in (**c**) showing the growth of ARK1ΔARM-GFP is not correlated with mCherry-RHD3.

ARK1–GFP comets/punctae moved together with an RHD3 punctum at the tip of an ER tubule (Fig. 9a, 0–19 s, arrows, Supplementary Movie 14. A large field view of Fig. 9a is provided in Supplementary Fig. 5). It pulled the ER tubule to grow until they met another ER tubule (Fig. 9a, 21 s, arrow, Supplementary Movie 14) where two ER tubules fused (Fig. 9a, 21 s, arrow, Supplementary Movie 14). This ARK1–GFP stayed with the newly formed junction of the ER and moved together for a while before both ARK1–GFP and mCherry-RHD3 signal disappeared (Fig. 9a, 21–28 s, arrows, Supplementary Movie 14). Kymograph analysis confirmed that ARK1–GFP moved together with RHD3 (Fig. 9b, A large field view of Fig. 9b is provided in Supplementary Fig. 6). Consistent with what was reported, ARK1ΔARM-GFP also marked plus ends of MTs[39] (Fig. 9c, arrows), in addition to punctae on MTs. Interestingly, of 30 ARK1ΔARM-GFP comets/punctae examined, none of them moved together with mCherry-RHD3 (Fig. 9c, arrows, Supplementary Movie 15). Kymograph analysis confirmed this notion (Fig. 9d). It has been shown that, in Arabidopsis hypocotyl epidermal cells, microtubule-dependent ER tubule elongation is slower than that of the actin–myosin system dependent[14]. Consistent with this, we found that ARK1-positive growing-end-dependent ER extension was at $0.099 ± 0.027$ μm/sec ($n = 28$), preexisting MT-dependent ER extension was at $0.415 ± 0.136$ μm/s ($n = 88$), but MT-independent ER extension was at $3.46 ± 1.15$ μm/s ($n = 88$).

## Discussion

Although it is generally believed that in plant cells, dynamics of the ER is largely driven by the actin–myosin system[8–11], limited evidence suggested that microtubules also play a role, in particular, in rapid growing cells[12,13]. In this study, we provide a further genetic evidence that MTs do play a role in the dynamics of the ER in plant cells. It is interesting to note that, in growing root hairs of *ark1*, the ER is defective, and in growing root hairs of *rhd3*, MTs are defective as well. Though it is possible that abnormal morphology of root hairs per se may also contribute to abnormal ER and MTs observed in the respective mutants. The data likely support the idea[13] that, in polarized growing root-hair cells, there is a bidirectional influence between MTs and the ER.

How could MTs contribute to ER dynamics in plant cells? It was reported by Hamada et al. that, in Arabidopsis hypocotyl epidermal cells, ER tubules can elongate along preexisting microtubules, but not at the growing tip of MTs[14]. We show here using ARK1–GFP that ER tubules can elongate along preexisting microtubules as well as at the growing ends of microtubules. Our quantification of the ratio of preexisting microtubule- and growing-end-dependent ER elongation suggests that the later contributes substantially to MT-dependent ER elongation in leaf epidermal cells. Likely, Hamada et al. missed the growing end-dependent ER elongation[14], as GFP-tubulin is not ideal to mark the growing ends of MTs or that the Lat-B treatment in their cells

may have an influence on the growing-end-dependent ER elongation.

How could the ER elongate with the growing end of MTs? We show here that, via the ARM domain, ARK1 physically interacts with RHD3, which is an ER fusogen. In leaf epidermal cells, RHD3 localizes to the ER with a concentration at the tip of growing ER as well as the three-way junction of the ER[19], while ARK1 localizes to the growing end of MTs as comets. We observed in leaf epidermal cells that ARK1–GFP comets could co-move with mCherry-RHD3 at the tip of an ER tubule and pull the ER tubule toward another, whereby two ER tubules become fused together. ARK1–GFP then stays with the newly formed junction of the ER and moves together for a while before both ARK1–GFP and mCherry-RHD3 signal disappeared. When the ARM domain of ARK1 is deleted, ARK1ΔARM is unable to move together with RHD3. Clearly, ARK1 and RHD3 can act together, such action may serve as an MT–ER contact site for an MT growing-end-dependent ER extension. In mammalian cells, EB1, a microtubule plus-end protein and STIM1, an ER-localized transmembrane protein bind and move together on the plus end of microtubules to lead extension of ER tubules through the TAC mechanism[5], yet, the motor for such movement is not studied. Here we show that, in plant cells, ARK1, a plant-specific armadillo-domain-containing kinesin binds and guides the movement of RHD3 on the plus end of microtubules so that an ER tubule can be extended until it is fused with another. Because ARK1 without the armadillo domain failed to interact and move with RHD3, it is highly likely that this direct interaction between ARK1 and RHD3 contributes at least in part to how ARK1 could regulate the ER morphology in plant cells. In mammalian cells, although there is no armadillo-domain kinesin, the action of some kinesins such as KIF3 is regulated by armadillo proteins[41]. Thus, it will be interesting to test if a similar mechanism exists in mammals. ARK1 is known to promote catastrophe of MTs[28], ark1 mutant plants exhibit a defect on the MT organization[28–30], as we have observed in ren9-1. Thus, it is highly possible that ARK1 also indirectly exerts its control over the ER through its regulation on MT organization. It is interesting to note that the expression of ARK1 without the armadillo domain rescues the dynamics of MTs[39]. It will be interesting to check if the expression level of ARK1ΔARM in ark1-1 is at the endogenous ARK1 level and then examine how well the ER will be restored by ARK1ΔARM in that ark1-1 line expressing ARK1ΔARM[39]. This would help us to figure out the contribution of the direct influence of ARK1 via RHD3 and the indirect influence of ARK1 via MTs on the ER.

In our study, ARK1 is predominantly expressed in growing root-hair cells and is mainly distributed as comets or punctae in front of MTs in the cortex of the subapical region of growing root hairs, while RHD3 is focused in the subapical region. Our 3D modeling of growing root hairs expressing gARK1–GFP and RFP-RHD3 indicates that these two proteins partially overlap in the cortex of the subapical region and are functionally interdependent. How important are this overlap and interdependence in ER dynamics and tip growth in root hairs? First, ER/RHD3 streaming in growing root hairs is only ~0.3 μm/s. Second, root hairs grow at a rate of ~0.015–0.02 μm/s[42]. In our quantification, MT-independent ER extension is at $3.46 \pm 1.15$ μm/s. MT-dependent ER extension is at $0.415 \pm 0.136$ μm/s, and ARK1-dependent ER extension is at $0.099 \pm 0.027$ μm/s. Thus, it would be difficult to count the fast actin–myosin-dependent ER extension as a major driven force for ER dynamics and tip growth observed in root hairs. It is likely in growing root hairs, MT-rather than actin-dependent ER extension plays a major role in ER dynamics. This view is supported by our results that RHD3 genetically interacts with MTs, but not actin[13]. We think that, in growing root hairs, the cortical microtubule lattice right behind

ARK1 in the subapical region would provide anchoring sites for ER tubules, which act as the branching sites of the ER[14]. ARK1 would then move ER tubules and also control where they should move in the cortex of subapical region, together with RHD3. The cortical fine ER network, once formed, would then provide a basis for ER extension into the endoplasm. Without ARK1, the movement of RHD3 is not well controlled, so RHD3 is seen in the shank region behind the subapical region, as well as in the apical dome.

Interestingly, in this study, RHD3 is also involved in the MT arrangement during tip growth. The interpretation of the result could be that RHD3 also influences the action of ARK1. In Drosophila, Atlastin interacts with Spastin to disassemble MTs in muscle cells[43]. Perhaps Arabidopsis RHD3 could also promote the action of ARK1 to disassemble MTs[28]. Failure to do so would lead to bundled MTs as we observed in growing root hairs of rhd3. However, it has been reported that the ARM domain of ARK1 is not required for microtubule catastrophe[39], and the interaction between ARK1 and RHD3 is through the ARM domain of ARK1; thus, the regulation of ARK1 by RHD3 through the interaction is less likely. In rhd3 mutant cells, the direction but not the velocity of the ARK1 movement is altered, it is possible that RHD3 could guide the direction of the ARK1 movement. During the growth of root hairs[13] and in epidermal leaf cells[44], ER tubules can occasionally co-align with MTs. In this study, ARK1 always moves in front of RHD3 in the cortex of the subapical region of growing root-hair cells, thus, the ER may serve as a supporting scaffold for the ARK1 movement. Finally, it is reported that, in xylem cells, an MT motor protein Kinesin-13A that depolymerizes MTs is activated by pit-localized ROP GTPase (ROP11)[45,46]. The action of ARK1 is under the regulation of ROP2[47], a ROP GTPase involved in the tip growth of root hairs[48]. In this regard, it is worth mentioning that the targeted transport of ROP2 to the apical dome requires RHD3[13]. Thus, RHD3 may also indirectly influence the action of ARK1 via ROP2.

## Methods

**Plant materials, growth conditions, and REN9 identification**. All plants were derived from the Columbia-0 (Col-0) ecotype unless mentioned. Seeds of rhd3-1 were treated with 0.3% EMS, and M2 seeds were harvested and screened[49]. ren9-1 and ren9-2 in the rhd3-1 background with the altered root-hair phenotype were screened from M2 seedlings. The observed phenotype was confirmed in the M3 generation seedlings. ren9-1 and ren9-2 in the rhd3-1 background were also backcrossed to rhd3-1 and wild-type Col-0 for the genetic segregation analysis. The root-hair length and bending angle related to the straight growth direction of root hairs of each genotype were measured and quantified with FiJi (http://fiji.sc/). For root-hair length, each root hair was traced using the "segmented line" tool in FiJi for measurement. For the root-hair bending angle, the angles were defined based on their bending directions. For wavy root hairs, bending angles to each direction were measured. For curly or helical root hairs that continuously bend in the same direction, the angle between the original growth direction and the final growth direction was measured. All angles were measured using the "angle tool" in FiJi. To identify the REN9 gene, ren9-1 was backcrossed to wild-type Landsberg. 52 F2 homozygous for ren9-1 were used to map the REN9 gene to chromosome 3. The genomic DNA of ARK1 in ren9-1 and ren9-2 was then amplified and sequenced.

For in vivo imaging of growing root hairs, seeds were sterilized and placed on the coverslip–agar system[50] covered by 1-mm thick AT (Arabidopsis thaliana) medium[49] supplemented with 1% sucrose. After cold-treated at 4 °C for 2–4 days, they were grown obliquely for 5–7 days at 22 °C under continuous light before imaging. For drug treatment, taxol purchased from Aldrich-Sigma (USA) was dissolved in DMSO and added to the medium at a final concentration of 1 μM.

**Plasmid construction and Arabidopsis transgenic lines**. The ARK1 genomic sequence (893 bp upstream of the ATG/start codon and coding region without TGA/stop codon) was amplified from the Arabidopsis genomic DNA with primers 5′-caccgcagtggaagaacac-3′ and 5′-gcttgagaagtaagggtttgttttg-3′, subcloned into vector pCR8 and then fused with GFP in pMDC107 by LR reaction to generate ARKpro:gARK1–GFP. To generate ARK1–GFP, cDNA of ARK1 was first cloned into the pGEM-gate entry vector and then subcloned into pEarleyGate 103. To generate YFP-RHD3, RFP-RHD3 or mCherry-RHD3, The GFP fragment of pVKH18-GFP-RHD3[19] was replaced by YFP, or RFP or mCherry with SalI and XbaI cutting sites. To generate YFP-Sey1p[682–776].36, the Sey1p[682–776] CDS region

was first cloned into pGEM-Gate entry vector and then subcloned into pEarleyGate 104. To generate mCherry-MAP4[38], the MBD domain of the mammalian MAP4 was cloned from mammalian CDS and fused with mCherry in its N-terminus in the pCambia1300 vector.

For stable transgenic lines, ARK1pro:gARK1–GFP was transformed to ren9-1 or ark1-1, YFP-RHD3 or RFP-RHD3 was transformed into rhd3-1 or rhd3-8, mCherry-MAP4[38] and YFP-Sey1p[682–766] [36] were introduced into Arabidopsis (Col-0), by the flower-dip method[51]. The Arabidopsis line of YFP-CLSD3[32] was obtained from Dr Erik Nielsen (Univ Michigan, USA) YFP-RAB-A2a[31] was obtained from Dr Ian Moore (Univ. of Oxford, UK). The rhd3 plants carrying ARK1pro:gARK1–GFP were segregated from the F2 of crossing offspring [rhd3-1 × ark1-1 (ARK1pro:gARK1–GFP)]. ren9-1 (ARK1pro:gARK1–GFP) and Col-0 (mCherry-MAP4) were crossed to co-express gARK1–GFP and mCherry-MAP4. ren9-1 (ARK1pro:gARK1-RFP and Col-0 (YFP-Sey1p[682–766]) were crossed to co-express gARK1-RFP and YFP-Sey1p[682–766]. ren9-1 (ARK1pro:gARK1–GFP) and rhd3-1 (RFP-RHD3) were crossed to co-express gARK1–GFP and RFP-RHD3. The F2 generation with wild-type root hairs were used for imaging.

**Living-cell imaging**. Living cell images were mainly acquired with a Leica SP8 point-scanning confocal system equipped with a ×63 oil objective and using 488 (GFP/YFP) and 552-nm (RFP/mCherry) lasers. Imaging of various reporter proteins during root-hair tip growth was carried out by a Quorum WaveFX spinning disk confocal system mounted on a Leica DMI6000B inverted microscope. A 491-nm laser line with a complementary GFP/YFP (520/35) emission or a 561-nm laser with a complementary RFP/mCherry (624/40) emission band-pass filter was used. For long-period imaging, images were acquired with a ×63 (oil) objective lens every 30 s (for YFP-CLSD3 and YFP-RAB-A2a) or 1–2 min (for YFP-Sey1p[682–766] and mCherry-MAP4) with the z-step size of 0.5–1 μm. For short-period image of YFP-CLSD3 and YFP-RAB-A2a, the acquiring interval was changed to 1 s. Short-period imaging of RFP-RHD3 and gARK1–GFP were done with a ×100 (oil) objective lens every 100 ms (RFP-RHD3) or 2 s (gARK1–GFP).

**Image process, analysis, and 3D modeling**. All images were processed and analyzed using ImageJ (http://fiji.sc/). For quantification of secretion swing in growing root hairs, Z-projection images (YFP-RAB-A2a) or optical sections at the midplane (YFP-CLSD3) were measured with the angle tool based on the previous growth direction of root hairs. RHD3 streaming was evaluated[11] by KbiFlow plug-in available through ImageJ (The package of KBI is download from http://hasezawa.ib.k.u-tokyo.ac.jp/zp/Kbi/ImageJKbiPlugins). Mask images for KbiFlow analysis were generated to remove the non-ER regions. The time-lapse sequence of 100 frames at a 512 × 512-pixel resolution was used in the analysis. The velocity of gARK1–GFP, ARK1–GFP, and mCherry-RHD3 was measured using MTrackJ plug-in. 3D modeling of gARK1–GFP/mCherry-MAP4 and gARK1–GFP/RFP-RHD3 was done with the Surface renderer in Huygens (svi.nl). Deconvolution of ARK1–GFP and RFP-RHD3 was performed with the Deconvolution Express in Huygens.

**Protein interaction assays**. To build a 3-in-1 BiFC vector cassette where mCherry-HDEL can be expressed from the same gene cassette, the P19 fragment of pDOE-03 vector[52] was replaced by the mCherry-HDEL fragment with KpnI-NruI cutting sites. Then RHD3 was fused with nVenus fragment at the N-terminus of RHD3, and ARK1, ARK1ΔARM and ARM of ARK1 were cloned and fused with cVenus fragment at their C-termini with AQUA cloning[53]. The transient expression of the construct was done with $OD_{600} = 0.01$.

For co-immunoprecipitation, ARK1–GFP, ARK1ΔARM-GFP, and GFP were co-expressed with mCherry-RHD3 in Nicotiana benthamiana leaves with $OD_{600} = 0.01$. Seventy-two hours after infiltration, infiltrated leaves were grinded into a powder in liquid nitrogen and extracted with the extraction buffer (50 mM Tris-HCl pH 7.5, 150 mM NaCl, and 10% (v/v) glycerol, and 0.5% IGEPAL® CA-630 (#I8896, sigma), 1/100 volume of Protease Inhibitor Cocktail (#P9599, sigma)). The solution was homogenized to an even mixture and centrifuged at $17,000 \times g$ for 10 min at 4 °C. Then the supernatant was collected. In total, 25 μl GFP-Trap®_MA beads (#gtma-20, Chromotek) were washed three times with the wash buffer (10 mM Tris-HCl pH 7.5; 150 mM NaCl). The protein lysate was added to the washed beads and incubated for 1 h at 4 °C. The beads were magnetically separated and washed three times with the wash buffer. Then the beads were resuspended in 50 μl 2× SDS-loading buffer and boiled for 10 min. The solution was centrifuged, and the supernatant was used for western blot. A rabbit anti-GFP antibody (Abcam, ab32146) at 1:5000 dilution or a rabbit anti-RFP (Abcam, ab34771) at 1:2000 dilution was used. The secondary anti-rabbit IgG-peroxidase (Sigma, A4914-1ML) was used at 1:5000 dilution.

**Reporting summary**. Further information on research design is available in the Nature Research Reporting Summary linked to this article.

## Data availability
Sequence data from this article can be found in the Arabidopsis Genome Initiative database under accession number At3g13870 (AtRHD3) and At3G54870 (AtARK1). Data supporting the findings of this study are provided within the paper and its supplementary files. All additional data are available from the corresponding author upon request. Source data are provided with this paper.

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

## Acknowledgements
We thank Dr. Natsumaro Kutsuna (University of Tokyo) for kind assistance in the quantitative analysis of RHD3 streaming, the Cell Imaging and Analysis Network (CIAN), and the Advanced Bio-Imaging Facility (ABIF) at McGill University for confocal microscopy imaging support; J.S and M.Z. were supported by a scholarship from the Chinese Scholarship Council. This research was supported by a discovery grant and Discovery Accelerator Supplement Award from the Natural Sciences and Engineering Research Council (NSERC) of Canada to H.Z.

## Author contributions
J.S. and M.Z. performed most of the experiments, analyzed the data; X.Q. performed the identification of ren9; C.D. performed the screen of ren9-1 and ren9-2. H.Z. designed the project and wrote the article with the contributions of other authors.

## Competing interests
The authors declare no competing interests.
