## [Peer Review File · Nature Communications]

Reviewers' comments:

Reviewer #1 (Remarks to the Author):

ARK1, a kinesin 1 protein acts with RHD3 to link microtubules to the ER for generation of a fine ER network in Arabidopsis

Jiaqi Sun, Mi Zhang, Xingyun Qi, Caitlin Doyle and Huanquan Zheng

This manuscript describes characterization of the role of an enhancer mutant called *ren9*. Mutation of *ren9* in an *rh3* background enhances root hair developmental defects. The authors describe mapping and confirmation of the *ren9* locus to a previously characterized kinesin motor protein, called ARK1. While ARK1 had previously been identified to cause root hair developmental defects, and displayed microtubule dynamics defects, in this manuscript the authors describe a potentially novel interaction between RHD3 and ARK1 which may affect ER dynamics, and adversely effect tip-restricted growth during root hair elongation. If sufficiently supported, this would uncover a novel role for RHD3, and provide important insight into mechanisms by which microtubules and ER membranes interact and how the placement of these ER membranes may be controlled during organization of the polarized subapical cytoplasm in tip-growing cells. However, there are a number of issues that need to be addressed before this manuscript would be suitable for acceptance to Nature Communications.

Main issues:

1. A major aspect of the authors' contention that ARK1 functions to control ER positioning in root hairs through interactions with the RHD3 protein are based on an apparent co-movement of ER-labeled membranes and/or ER-associated RHD3 with fluorescently tagged ARK1. Unfortunately the quality of the current images and movies provided makes it difficult to interpret this contention.
 - a) In Figure 3, the authors utilize a YFP-Sey1p(682-766) fusion to apparently label ER membranes in root hairs. I don't know why they are using this rather than more commonly used ER markers, such as GFP-HDEL, or GFP-Calreticulin? The reason I am a bit worried about this yeast-derived marker is that it's distribution in the root hairs shown in figure 4C looks quite filamentous in nature. This doesn't really look like most ER-marker localizations in tip-growing cells. They clearly believe they would likely see colocalization with an mCherry-HDEL in figure 6B, so it would be important to use a more common ER marker for these ARK1/ER colocalizations in figure 4.
 - b) In figure 4A-B they clearly establish that, as previously reported, ARK1 colocalizes with MT ends at a sub-apical domain (presumably behind the vesicle-rich zone), but the colocalization between fluorescently-labeled ARK1 with either YFP-Sey1p (fig 4C-D) or RFP-RHD3 are not as clear. While the placement of the ARK1-GFP is clearly proximal to YFP-Sey1p and RFP-RHD3 labeled compartments, this would probably be more clearly represented in a 3D-movie from a z-stack. However, in neither case is the proximal localization of the ARK1-GFP causal for localization of ER-labeled membranes. This is particularly apparent in figure 5B and supplemental movie 4 where ARK1-GFP heavily labels MT ends, but these are not found associated with concomitant enrichment of RFP-RHD3 labeled membranes. While the authors attempt to alter MT dynamics (and therefore ARK1-GFP localization) with taxol in figure 5, the RFP-RHD3 redistribution is not necessarily correlated with altered ER, and the dynamics of ARK1-GFP and RFP-RHD3 fluorescence is not convincing. I see a lot of bundled ARK1-GFP dynamics, but these are not associated with significant RFP-RHD3 labeled structures. It might be helpful for the authors to look at ER-membrane markers (other than YFP-Sey1p) and RFP-RHD3 dynamics in the *ren9-1* mutant background. Altered positioning or dynamics of ER-membranes or RFP-RHD3 in this background might more strongly support the causal nature of a potential RHD3-ARK1 interaction on ER dynamics.
2. The in vitro co-purification and in vivo BiFC labeling experiments that the authors present to support the physical interaction of ARK1-ARM domains with RHD3 are pretty rudimentary and require important additional controls.
 - a) In figure 6A, the authors describe co-immunoprecipitation of transiently-expressed RFP-RHD3 with ARK1-GFP, but not GFP alone. One problem with this experiment is that it is not clear how

specific these co-precipitations are. While GFP is likely to remain a soluble protein, it is unclear how soluble ARK1-GFP is, or whether it may aggregate into higher-order complexes (either alone, or with microtubules) which might trap the RFP-RHD3. It would be useful to either i) use an additional kinesin which is not predicted to recruit RHD3, or ii) show that presence of ARK1-GFP can recruit RHD3 to stabilized microtubules in a specific manner.

b) In figure 6B, the BiFC results are not convincing. First, the contrast in these images is far too high, which likely masks faint BiFC signals. Second, why is the colocalization of BiFC signal from the ARK1-RHD3 (or ARM domains) not observed consistently at ER-tubule junctions? The incredibly sparse presence of the BiFC signal within these cells makes it difficult to interpret whether this reconstituted fluorescence is due to functional protein complexes. Also, why is there significant BiFC fluorescence on the limiting membranes of these cells? Is it possible that this reflects a significant recruitment of these proteins to plasma membranes?

Other issues:

1. In figure S1 the authors present evidence that placement of Rab-A2a labeled compartments changes during the wavy or curvy growth phenotype of root hairs in the *rhd3* mutant background. This altered positioning has previously been described for Rab-A4b (Thole et al., 2008), and was proposed to be due to altered positioning of the tip-localized expansion zone (Galway et al., 1997). These observations should be noted when discussing these results.

2. In figure S5 the authors note apparent differences in the organization of ARK1-GFP on microtubules in some root hairs. Rather than a consistent "ring" of labeling at the flanks of the growing tips of these cells, they instead describe localization to the extreme apex of some cells. Are these cells still undergoing tip growth? It seems likely that they are observing cells that have stopped tip-restricted elongation, and that this extension of ARK1-GFP labeled microtubules simply reflects the loss of the vesicle-rich zone in these cells.

3. Finally, this article contains a multitude of spelling and grammatical mistakes (too numerous to list them all). The authors are encouraged to seek out help with editing future versions of this manuscript.

Reviewer #2 (Remarks to the Author):

Dear authors and editors,

This article by Sun et al. demonstrates functional interaction between armadillo-repeat kinesin (ARK1) and dynamin-like atlastin GTPase (RHD3) in root hair growth, providing a proxy of interaction between microtubules (MTs) and endoplasmic reticulum (ER).

ARK1 promotes microtubule catastrophe to enhance microtubule dynamics and polymerization. RHD3 is an ER fusogen promoting ER network formation. Both are required for tip growth of root hairs, but their interaction remains unknown. Genetic modifier screening identifies *ark1* mutations as an enhancer of *rhd3* mutant. Both *ark1* and *rhd3* mutants exhibit defects in the organization of ER and microtubules. The double mutant exhibits severe defects in ER, microtubules, and directional secretion detected by labelling of RABA2 and CLSD3. ARK1 localizes in the leading edge of subapical ER clouds but not colocalize with RHD3 in root hairs. Taxol treatment suggests microtubule-dependent RHD3 movement. ARK1 and RHD3 physically interact in IP and BiFC. ARK1 localizes in the tip of RHD3-positive ER tubules and remains till ER fusion in the transient assay in tobacco leaf cells.

The results support microtubule-dependent elongation of ER tubules in plants, previously described in reference 12-14 but still underestimated. This mechanism may specifically function in rapid cell growth and cooperate with the actomyosin-mediated ER movement during cytoplasmic streaming (ref. 9-11).

The novelty is the mode of interaction of microtubule-ER via ARK1-RHD3 module: ARK1 kinesin motor guides the elongation of ER tubules along microtubules to promote their fusion and network formation. It is interesting because motor protein responsible for ER elongation along microtubules still remains unknown. However, this version of manuscript is still descriptive and is not fully convincing, so I recommend quantification of results, additional experiments, and reorganization of figures.

Comments

1. The requirement and functional significance of ARK1-RHD3 interaction for root hair growth remain obscure. The ARM domain of ARK1 is dispensable for tip growth of root hairs (Eng et al. 2017), whereas ARM is essential for the interaction and co-movement of ARK1 and RHD3.
2. It is not convincing whether *ren9* and *rhd3* mutation are additive or synergistic. Quantification of length and waviness of root hairs in single and double mutants is helpful to support synergistic interaction of ARK1 and RHD3.
3. Because supplementary Fig S1 and S2 are very clear and significant to show functional interaction of ARK1 and RHD3 in root hairs, I recommend moving them (or one of them) into main figures. To make the space for that, Fig 1 and 2 could be combined (Fig 2, especially Fig 2b and 2c, are not essential and can be supplementary figure).
4. Fig 3 is descriptive and qualitative. Quantification of ER signal in concave and convex sides is required.
5. Fig S4 clearly shows microtubule defect in *rhd3* and *rhd3 ren9* mutants. This is crucial for their functional interaction in microtubule regulation. Fig 3 and Fig S4 could be combined to demonstrate mutual dependence of ER and MT. Quantification of MT signal in concave and convex sides is also required.
6. ER defect in the *ark1* mutant and MT defect in the *rhd3* mutant might be caused by indirect effect of root-hair growth abnormalities in these mutants: growth defects induce disorganization of ER and MTs. Please explain and discuss this possibility.
7. Localization pattern of ARK1-GFP (such as absence in the apical dome) is not coincident with the results in Eng and Wasteneys (2014) and Eng et al (2017). Please provide reason and explanation for it.
8. Add a kymograph of ARK1-GFP and mCherry-MAP4 in Fig 4.
9. In Fig 4 and Fig 5, ARK1 does not colocalize with Sey1p-labelled ER and RHD3, which mainly localize in the cytosol. ARK1 associates with cortical region, whereas ER network is formed in the endoplasm. Please explain this discrepancy.
10. If ARK1-RHD3 interaction in cell cortex is important, it is essential to show dynamics of ARK1-RHD3 in the cortex.
11. The purpose and conclusion of taxol treatment (line 214-224, Fig 5) and of effect of *rhd3* mutation on ARK1 (line 224-230, Fig S5) is obscure. It is better to separate the paragraph into two. Please add the rationale/purpose of experiments and make the conclusion clear. Fig S5, especially Fig S5a is important, so it should be moved into main figure (a part of Fig 5).
12. I wonder whether the *ren9/ark1* mutation affects localization of RHD3. If ARK1 guide and/or recruits RHD3, its localization and dynamics are affected in *ren9/ark1*.

13. In BiFC (Fig 6b), RHD3-ARK1 signal associates but not colocalize with ER, whereas RHD3-ARM signal colocalizes with ER. Please explain the reason for it.

14. I wonder why the colocalization pattern of RHD3 and ARK1 in Fig 7 is quite different from BiFC signal pattern of RHD3-ARK1 in Fig 6. Please explain it and provide larger field images of ARK1-GFP plus mCherry-RHD3 (Fig 7a) and ARK1 dARM-GFP plus mCherry-RHD3 (Fig 7b).

15. Because Fig 7 is too much descriptive and qualitative, please quantify growth speed and run distance of ARK1-RHD3-positive ER tubules. In addition, kymographs are required in ARK1 positive/negative ER tubules. Microtubule-dependent ER extension is 20 fold slower than the actin-myosin mode, so the growth speed will provide the evidence which system works on.

16. In Arabidopsis, ER tubules do not extend with growing microtubule ends but rather elongate along preexisting microtubules toward both plus and minus ends (Hamada et al. 2014). ARK1 localizes in growing plus ends of microtubules, so the result in Fig 7 indicates a new mode: ER elongation with growing plus ends. Please explain the discrepancy and provide quantification of the ratio of ARK1-positive ER extension events (growing-end dependent) versus ARK1-negative ER extension events (preexisting-microtubule-lattice dependent). This quantification will provide the contribution of ARK1-dependent ER extension mechanism in the whole ER network and elongation. The frequency of ER fusion events per encounter with other ER (in both ARK1-positive/negative ER ends) will be helpful to clarify functional significance of ARK1-dependent mechanism.

17. Cortical microtubule lattice provides the anchoring sites of ER tubules, which act as the branching sites of ER (Hamada et al. 2014). The plus-end ARK1-dependent extension promotes encounter of ER ends with other ER structure, and RHD3 physically interacted with ARK1 induces ER fusion. The former mechanism increases ER network complexity and ER free ends, whereas the latter reduces complexity and free ends. I wonder functional significance of ARK1-dependent extension mechanism in root hair growth (how this mechanism contributes ER organization, especially in subapical ER network of root hairs). And why is ARK1 used for ER extension-fusion mechanism?

18. Microtubule-stabilizing compound, taxol, promotes ark1 phenotype, whereas microtubule-depolymerizing compound, oryzalin, partially rescues ark1 (Eng and Wasteney 2014). This supports that ARK1 increases microtubule catastrophe and free tubulin concentration to enhance new microtubule elongation. However, both taxol and oryzalin partially rescue rhd3 phenotype (Qi et al. 2016). Please explain the discrepancy of inhibitor effects.

Reviewer #3 (Remarks to the Author):

Review for: Sun et al. "ARK1, a kinesin protein acts with RHD3 to link microtubules to the ER for generation of a fine ER network in Arabidopsis"

In this manuscript the authors state that the interaction between the Armadillo Repeat Kinesin1 protein (ARK1) and RHD3 leads to tubule fusion and the formation of a fine ER network. The authors suggest that this is due to the proteins linking microtubules and ER.

Whilst this is an intriguing possibility and highly interesting for the scientific communities some of the data and especially the imaging is letting the work down and should be improved before publication. In addition statistical image analysis should be applied to show co-localisations or structural changes rather than only showing single images or image sequences.

Figure 4b: the co-localisation here is really hardly visible and the movie is very fussy. The mCherry-ARK fusion seems to show less comets/punctae than a GFP version?

Figure 4c requires higher resolution

Figure 5 a,b: again here I struggle to see how ARK1 is surrounding RHD3 due to a lack of resolution.

Figure 5d: I cannot see a real co-localisation of the ARK1 bundles with RHD3, rather the opposite in at least two cases where a bundle of ARK1 has no RHD3 label

Figure 6a: the Western blot background should not be removed with the bands exaggerated. The size marker should be on the same gel or sizes can be indicated by labeling.

Figure 6b: I cannot see any clear ER labeling with any of the combinations. This figure is insufficient for publication.

Figure 7: I appreciate that imaging in root hair cells has its difficulties but this is now done in tobacco leaves and therefore the imaging resolution can be improved. Especially (but not only) in a printed version the images are very unclear and pixelated.

Although the ARM deletion shows no interactions, the authors should comment why only 10 out of 30 co-move. I can also not find how many biological and technical replicas were used.

The ARK1 and Δ ARM labelling looks rather different. Is that representative?

Figure S1 is lacking labelling a-e

Also the dataset is not fully clear to me and would really need a better and more in-depth figure legend

Figure S4 should be combined with Figure 3.

Line 76: here Sparkes et al 2010 "Five Arabidopsis Reticulon Isoforms Share Endoplasmic Reticulum Location, Topology, and Membrane-Shaping Properties" should be cited for the RTN topology investigation

Line 239: the statement that an OD of 0.01 is not considered overexpression is not correct. I appreciate that overexpression is being a difficult terminology but given that there is additional expression of a non-native protein this statement is misleading and should be removed.

Line 631 "of plants indicated": please clarify the title

Point-by-point response to reviewers' comments:

Reviewer #1 (Remarks to the Author):

.....

Main issues:

1. A major aspect of the authors' contention that ARK1 functions to control ER positioning in root hairs through interactions with the RHD3 protein are based on an apparent co-movement of ER-labeled membranes and/or ER-associated RHD3 with fluorescently tagged ARK1. Unfortunately the quality of the current images and movies provided makes it difficult to interpret this contention.

a) In Figure 3, the authors utilize a YFP-Sey1p(682-766) fusion to apparently label ER membranes in root hairs. I don't know why they are using this rather than more commonly used ER markers, such as GFP-HDEL, or GFP-Calreticulin? The reason I am a bit worried about this yeast-derived marker is that its distribution in the root hairs shown in figure 4C looks quite filamentous in nature. This doesn't really look like most ER-marker localizations in tip-growing cells. They clearly believe they would likely see colocalization with an mCherry-HDEL in figure 6B, so it would be important to use a more common ER marker for these ARK1/ER colocalizations in figure 4.

We appreciate this constructive criticism. Although GFP-HDEL is a commonly used marker for the ER in epidermal cells of leaves and hypocotyls, it is not a good marker to mark the ER in rapidly growing root hairs (Chen et al., 2012; Qi et al., 2016). It has been reported that in growing root hair tips, there is a fine ER focus in the subapical region (Chen et al., 2012; Galway et al., 1997; Qi et al., 2016). This fine ER cannot be labelled properly by luminal GFP-HDEL (Chen et al., 2012; Qi et al., 2016), which produces aggregates in the tip region of growing root hairs and also has a negative influence in root hair growth (Chen et al., 2012), likely due to ER stress induced by the expression of GFP-HDEL in the ER lumen. However, we indeed revealed that some ER membrane proteins, such as RHD3, HVA22d (Qi et al., 2016) and Sey1p682-766 (Stefano and Brandizzi, 2014) can be used to properly label the fine ER in growing root hairs. Because HVA22d labels some Golgi structure (also have some negative effect on tip growth) (Qi et al., 2016) and RHD3 cannot be used for ER observation in *rh3* mutants, We choose to use YFP-Sey1p682-766 to label the ER in growing root hairs. The expression of YFP-Sey1p682-766 does not rescue *rh3* and there is no negative influence on root hair growth, thus it is an ideal marker for ER imaging in growing root hairs.

b) In figure 4A-B they clearly establish that, as previously reported, ARK1 colocalizes with MT ends at a sub-apical domain (presumably behind the vesicle-rich zone), but the colocalization between fluorescently-labeled ARK1 with either YFP-Sey1p (fig 4C-D) or RFP-RHD3 are not as clear. While the placement of the ARK1-GFP is clearly proximal to YFP-Sey1p and RFP-RHD3 labeled compartments, this would probably be more clearly represented in a 3D-movie from a z-

stack. However, in neither case is the proximal localization of the ARK1-GFP causal for localization of ER-labeled membranes. This is particularly apparent in figure 5B and supplemental movie 4 where ARK1-GFP heavily labels MT ends, but these are not found associated with concomitant enrichment of RFP-RHD3 labeled membranes.

As the reviewer #3 pointed out, ARK1-RFP is not as good as ARK1-GFP in labeling plus end of microtubules, so in this revised manuscript, we used ARK1-GFP not ARK1-RFP to track the movement of ARK1. To better illustrate the spatial relationship between ARK1-GFP and mCherry-MAP4 as well as between ARK1-GFP and RFP-RHD3, we constructed a 3D-movie from a z-stack in each case. As it can be seen from Figure 5c and 6d and supplemental movies 2-4 and 7-9, while ARK1-GFP is enriched in the cortical region and localized in the plus end of microtubules (Figure 5c, supplemental Movies 2-4) in the subapical region, RFP-RHD3 is distributed in the endoplasm and also in the cortical region (Figure 6d, supplemental movies 7-9). Clearly, at least a portion of cortical ARK1-GFP overlaps with subapical focused RFP-RHD3 in the cortical region. This confirmed what we reported in Figure 6b and supplemental Movie 6.

While the authors attempt to alter MT dynamics (and therefore ARK1-GFP localization) with taxol in figure 5, the RFP-RHD3 redistribution is not necessarily correlated with altered ER, and the dynamics of ARK1-GFP and RFP-RHD3 fluorescence is not convincing. I see a lot of bundled ARK1-GFP dynamics, but these are not associated with significant RFP-RHD3 labeled structures.

The partial association of bundled ARK1-GFP with RFP-RHD3 in the endoplasm of taxol (1 μ M) treated cells is relative to ARK1-GFP and RFP-RHD3 in non-treated root hairs. After the Taxol treatment, we revealed an increased association of ARK1-GFP and RFP-RHD3 (yes, it is still partial), while in non-Taxol treated samples, the association is hardly found, if it is all possible. We have included this in supplemental figure S3.

It might be helpful for the authors to look at ER-membrane markers (other than YFP-Sey1p) and RFP-RHD3 dynamics in the *ren9-1* mutant background. Altered positioning or dynamics of ER-membranes or RFP-RHD3 in this background might more strongly support the causal nature of a potential RHD3-ARK1 interaction on ER dynamics.

We appreciate the comment. Yes, as the reviewer #2 also pointed out, it is better to examine the distribution and dynamics of RHD3 in *ren9-1* (rather than the taxol treatment to support the causal nature of the ARK1-RHD3 interaction). We have done so and revealed that YFP-RHD3 is no longer restricted in the subapical region and there are bundles in the endoplasm. We also found that the motility of RHD3 streaming is reduced in *ren9-1*. We have added this in Figure 7a-d and discussed this result in the discussion.

2. The in vitro co-purification and in vivo BiFC labeling experiments that the authors present to support the physical interaction of ARK1-ARM domains with RHD3 are pretty rudimentary and require important additional controls.

a) In figure 6A, the authors describe co-immunoprecipitation of transiently-expressed RFP-RHD3 with ARK1-GFP, but not GFP alone. One problem with this experiment is that it is not clear how specific these co-precipitations are. While GFP is likely to remain a soluble protein, it is unclear how soluble ARK1-GFP is, or whether it may aggregate into higher-order complexes (either alone, or with microtubules) which might trap the RFP-RHD3. It would be useful to either i) use an additional kinesin which is not predicted to recruit RHD3, or ii) show that presence of ARK1-GFP can recruit RHD3 to stabilized microtubules in a specific manner.

We agree that a negative control is needed. It is difficult to predict which plus end kinesin does not interact with RHD3 that can serve as a negative control. At the same time, ARK1 Δ ARM is known to also localize to microtubule plus-end and it even rescues microtubule dynamics (Eng et al., 2017), so we decided that to use ARK1 Δ ARM as a control in our Co-IP. We have added the result in Figure 8a.

b) In figure 6B, the BiFC results are not convincing. First, the contrast in these images is far too high, which likely masks faint BiFC signals.

Second, why is the colocalization of BiFC signal from the ARK1-RHD3 (or ARM domains) not observed consistently at ER-tubule junctions? The incredibly sparse presence of the BiFC signal within these cells makes it difficult to interpret whether this reconstituted fluorescence is due to functional protein complexes. Also, why is there significant BiFC fluorescence on the limiting membranes of these cells? Is it possible that this reflects a significant recruitment of these proteins to plasma membranes?

First, those signals that outlined membranes are auto-fluorescence. To avoid confusion, we have repeated our BiFC experiments with careful control of OD600. A new set of BiFC with lower contrast is presented in Figure 8b. As it can be seen, most ARK1-RHD3 BiFC signals are on either 3-way junctions or 2-way junctions of the ER (which appear to be on ER tubules), or at the tip of ER tubules. We have described this in the result. In our repeated experiments, it is very rare to see ARK1-RHD3 BiFC on ER tubules. We think that even if some ARK1-RHD3 signals are indeed on ER tubules, they may represent the points where ER tubules may grow.

Other issues:

1. In figure S1 the authors present evidence that placement of Rab-A2a labeled compartments changes during the wavy or curly growth phenotype of root hairs in the *rhd3* mutant background. This altered positioning has previously been described for Rab-A4b (Thole et al., 2008), and was proposed to be due to altered positioning of the tip-localized expansion zone (Galway et al., 1997). These observations should be noted when discussing these results.

This possibility is added in the text.

2. In figure S5 the authors note apparent differences in the organization of ARK1-GFP on microtubules in some root hairs. Rather than a consistent “ring” of labeling at the flanks of the growing tips of these cells, they instead describe localization to the extreme apex of some cells. Are these cells still undergoing tip growth? It seems likely that they are observing cells that have stopped tip-restricted elongation, and that this extension of ARK1-GFP labeled microtubules simply reflects the loss of the vesicle-rich zone in these cells.

We agree to this point that those cells have stopped tip-restricted elongation. The slight swollen morphology at that moment supports this view that, root hairs ceased its tip restricted elongation when ARK1-GFP is at the extreme apex. We have added this view in the manuscript.

3. Finally, this article contains a multitude of spelling and grammatical mistakes (too numerous to list them all). The authors are encouraged to seek out help with editing future versions of this manuscript.

We have checked this throughout the revised manuscript.

Reviewer #2 (Remarks to the Author):

Dear authors and editors,

.....

Comments

1. The requirement and functional significance of ARK1-RHD3 interaction for root hair growth remain obscure. The ARM domain of ARK1 is dispensable for tip growth of root hairs (Eng et al. 2017), whereas ARM is essential for the interaction and co-movement of ARK1 and RHD3.

Yes, the ARM domain appears dispensable for MT dynamics based on the quantified velocity of ARK1-GFP and ARK1 Δ ARM-GFP and microtubule plus end catastrophe (Eng et al., 2017). We do not think our data that the domain is important for the interaction and co-movement of ARK1 and RHD3 contradicts this published result. Here we like to point out that, first, it is not clear if the expression level of ARK1 Δ ARM-GFP in *ark1-1* in Eng et al. (2017) is at the level of endogenous ARK1. Second, we do not know how well the ER is restored by ARK1 Δ ARM in *ark1-1*. In this regard, we noted that root hairs of *ark1-1* expressing ARK1 Δ ARM-GFP appear thicker than wild type (Eng et al., 2017), implying that it is possible that the growth of root hairs were not fully rescued. We think that, in addition to its action on MT catastrophe, ARK1 acts

also as a motor protein to influence the movement of its cargoes (e.g. RHD3 and the ER in this case). Likely, ARK1 will influence the ER at least in two folds: first, directly through its interaction with RHD3 via ARM to elongate ER tubules (this study), second, indirectly through its action on MT dynamics as it has been shown that, ER tubules can elongate along preexisting microtubules toward both plus and minus ends (Hamada et al., 2014). We have added this view in the revised manuscript.

2. It is not convincing whether *ren9* and *rhd3* mutation are additive or synergistic. Quantification of length and waviness of root hairs in single and double mutants is helpful to support synergistic interaction of ARK1 and RHD3.

We thank the reviewer 2 for this thoughtful suggestion. We have added the quantification of the length and bending angle of root hairs in 4 different genotypes. The data is presented in Figure 1k-l.

3. Because supplementary Fig S1 and S2 are very clear and significant to show functional interaction of ARK1 and RHD3 in root hairs, I recommend moving them (or one of them) into main figures. To make the space for that, Fig 1 and 2 could be combined (Fig 2, especially Fig 2b and 2c, are not essential and can be supplementary figure).

We appreciate this comment. We have moved Figure S1 as Figure 2, and Figure S2 as Figure S1. As these results are described before the cloning of *REN9* (original Figure 2) so we moved this original Figure 2 as Figure 3. We think it is not proper to combine Figure 2 into Figure 1.

4. Fig 3 is descriptive and qualitative. Quantification of ER signal in concave and convex sides is required.

This is a good suggestion. We have quantified the distribution of ER by quantifying the signal intensity in a horizontal line crossing concave and convex in root hairs.

5. Fig S4 clearly shows microtubule defect in *rhd3* and *rhd3 ren9* mutants. This is crucial for their functional interaction in microtubule regulation. Fig 3 and Fig S4 could be combined to demonstrate mutual dependence of ER and MT. Quantification of MT signal in concave and convex sides is also required.

We have combined Fig S4 with Figure 3 as Figure 4. MT signal in concave and convex sides is also quantified.

6. ER defect in the *ark1* mutant and MT defect in the *rhd3* mutant might be caused by indirect effect of root-hair growth abnormalities in these mutants: growth defects induce disorganization of ER and MTs. Please explain and discuss this possibility.

We fully agree this point. It is possible that there is such an additional effect on ER and MTs from abnormal root hairs. We have added this possibility in the discussion in the revised manuscript.

7. Localization pattern of ARK1-GFP (such as absence in the apical dome) is not coincident with the results in Eng and Wasteneys (2014) and Eng et al (2017). Please provide reason and explanation for it.

We thank the reviewer 2 for pointing out this discrepancy. We in fact also noted that the localization pattern is different from Eng and Wasteneys (2014) (Eng and Wasteneys, 2014) and Eng et al. (2017) (Eng et al., 2017). We think that the difference may raise from different growth ages of cells examined, and/or from different expression levels between transgenic lines used. We have added this view into the revised manuscript. In our line, ARK1-GFP is predominantly presented in growing root hairs, whereas in Eng and Wasteneys (2014), ARK1-GFP is observable not only in growing root hairs, but also in other tissues, suggesting that there may be a difference in the expression of ARK1-GFP in lines used. Also, according to Van Bruaene (Van Bruaene et al., 2004), in the growing root hairs, MTs do not reach the very tip. But in full-grown or growth arresting root hairs, MTs reach the very tip of root hairs.

8. Add a kymograph of ARK1-GFP and mCherry-MAP4 in Fig 4.

We tried to add a kymograph but found out that it is very difficult to select an ARK1-GFP comet to follow its movement with mCherry-MAP4 at growing root hair tips. We instead created a 3D movie (Figure 5c and supplemental Movies 2-4) to better illustrate the spatial relationship between ARK1-GFP and cCherry-MAP4.

9. In Fig 4 and Fig 5, ARK1 does not colocalize with Sey1p-labelled ER and RHD3, which mainly localize in the cytosol. ARK1 associates with cortical region, whereas ER network is formed in the endoplasm. Please explain this discrepancy.

We have studied the spatial relationship between ARK1-RHD3 in detail in 3D. As it can be seen from Figure 6d and supplemental movies 7-9, while ARK1-GFP is enriched in the cortical region and localized at the plus end of microtubules (Figure 5c, supplemental Movies 2-4), RFP-RHD3 is distributed in the endoplasm as well as in the cortical region. Clearly, ARK1-GFP partially overlaps with subapical focused RFP-RHD3 in the cortical region. This confirmed what we reported in Figure 6b and supplemental Movie 6.

10. If ARK1-RHD3 interaction in cell cortex is important, it is essential to show dynamics of ARK1-RHD3 in the cortex.

The added 3D analysis in Figure 6d and supplemental Movies 7-9 clearly demonstrated the spatial relationship of ARK1-RHD3 in the cortex of the subapical region. To show dynamics of

ARK1-RHD3 in the cortex in 3D, it requires a production of series of 3D and then make a movie with the generated 3Ds. This is technically challenging for us right now. However, we believe our supplemental Movie 6 did show the co-movement of ARK1-RHD3 in the cortex of a growing root hair.

11. The purpose and conclusion of taxol treatment (line 214-224, Fig 5) and of effect of *rhd3* mutation on ARK1 (line 224-230, Fig S5) is obscure. It is better to separate the paragraph into two. Please add the rationale/purpose of experiments and make the conclusion clear. Fig S5, especially Fig S5a is important, so it should be moved into main figure (a part of Fig 5).

We appreciate this good suggestion. The purpose of the taxol treatment is to illustrate the causal relationship between ARK1 and RHD3. We agree that the better way is to study dynamics of RHD3 in the *ren9* mutant, as the comment #12 suggests. We have analyzed the distribution and dynamics of YFP-RHD3 in *ren9-1*. See our response to the comment #12 for the new result. We have combined the new results with original Fig S5 as a new Figure 7, and moved the taxol treatment into supplemental Figure S3.

12. I wonder whether the *ren9/ark1* mutation affects localization of RHD3. If ARK1 guide and/or recruits RHD3, its localization and dynamics are affected in *ren9/ark1*.

We fully agree with the reviewer 2 on this comment. We examined the distribution and dynamics of YFP-RHD3 in the *ren9-1* mutant background. We revealed that focused YFP-RHD3 is no longer restricted in the subapical region and there are bundles in the endoplasm. We also revealed that the motility of RHD3 streaming is reduced in *ren9-1*. In addition, the taxol (1 μ M) treatment mimics what is observed in *ren9-1* (Figure S3). We have added this in Figure 7a-d and Figure S3 and discussed the result in the discussion.

13. In BiFC (Fig 6b), RHD3-ARK1 signal associates but not colocalize with ER, whereas RHD3-ARM signal colocalizes with ER. Please explain the reason for it.

Yes, we also observed this: RHD3-ARK1 signal associates but not colocalizes with ER, whereas RHD3-ARM signal colocalizes with ER. We think that, with the motor domain, the full length of ARK1 is stably localized at the plus end of MTs, so the BiFC may indicated its labeling of ER-MT junctions, while without the motor domain, the association of ARM with MTs may be weakened, so the ARM domain can be brought to the ER by RHD3. We have added this view in the revised manuscript.

14. I wonder why the colocalization pattern of RHD3 and ARK1 in Fig 7 is quite different from BiFC signal pattern of RHD3-ARK1 in Fig 6. Please explain it and provide larger field images of ARK1-GFP plus mCherry-RHD3 (Fig 7a) and ARK1 Δ ARM-GFP plus mCherry-RHD3 (Fig 7b).

Original Figure 7 (now Figure 9) focuses on co-movement of ARK1-RHD3 at a tip of an ER tubule. Original Figure 6 (now Figure 8) focuses on BiFC signal between the two proteins. To

quantify the elongation of ER tubules, we paid our attention to the growing tip of the ER when the co-movement of ARK1-RHD3 punctae was studied, though ARK1-RHD3 punctae were also observed in ER junctions, for example, in 21-27' in Figure 7 (now figure 8). On the other hand, BiFC signal of ARK1-RHD3 can also be observed at the tip of ER tubules, in addition to junctions of the ER. We have replaced the old BiFC result with a new set of BiFC where BiFC can also be seen at the tip of an ER tubule in new Figure 8. We agree that there is a difference in the frequency between the co-localization of RHD3 and ARK1 punctae and BiFC signal of ARK1-RHD3. We think this may be a reflection of the fact that BiFC, once formed, tends to be stable (Robida and Kerppola, 2009), so BiFC signals of ARK1-RHD3 tend to accumulate in cells even after 2- or 3-way junctions are formed. The co-movement of ARK1-RHD3 punctae may be more close to a true dynamic situation where ARK1-RHD3 punctae would disappear after a while once 2- or 3- way junctions are formed (28' in Figure 7, now Figure 8).

Finally, we have also provided larger field images for ARK1-GFP plus mCherry-RHD3 (Fig 7a, now figure 9a) and ARK1 Δ ARM-GFP plus mCherry-RHD3 (Fig 7b, now Figure 9b) as supplemental Fig S4 and S5 respectively. Additional ARK1-RHD3 punctae can be seen.

15. Because Fig 7 is too much descriptive and qualitative, please quantify growth speed and run distance of ARK1-RHD3-positive ER tubules. In addition, kymographs are required in ARK1 positive/negative ER tubules. Microtubule-dependent ER extension is 20 fold slower than the actin-myosin mode, so the growth speed will provide the evidence which system works on.

We have quantified the velocity of ARK1-dependent tip growth of the ER ($0.099 \pm 0.027 \mu\text{m}/\text{sec}$, $n=26$), MT-dependent ER extension ($0.415 \pm 0.136 \mu\text{m}/\text{sec}$, $n=88$) and MT-independent tip growth of the ER ($3.46 \pm 1.16 \mu\text{m}/\text{sec}$, $n=89$). We have also provided a kymograph for ARK1-RHD3 as well as for ARK1 Δ ARM-RHD3 in new Figure 9.

Our ARK1-dependent tip growth rate of the ER is consistent with microtubule plus-end growth velocity revealed by ARK1-RFP in Arabidopsis leaf epidermal cells ($0.053 \pm 0.03 \mu\text{m}/\text{sec}$) and by EB1b-GFP in hypocotyls ($0.083 \pm 0.046 \mu\text{m}/\text{sec}$) (Eng and Wasteneys, 2014). Clearly, microtubule-dependent or ARK1-dependent ER extension is slower than the MT-independent, likely actin-myosin dependent mode. Further, we also quantified the ratio of MT- and ARK1-dependent and -independent ER elongation. We found that in mature tobacco leaf epidermal cells, the majority of ER extension events (89/91) are MT- or ARK1-independent. This suggested that, similar to what is observed in hypocotyl epidermal cells (Hamada et al., 2014), in tobacco leaf epidermal cells, MT-independent ER extension may play a major role in ER dynamics, which is supplemented by MT-dependent ER extension. However, we think it is unlikely that, the fast actin-myosin dependent mechanism will play a major role in ER streaming in growing root hairs. First, as we revealed in this study, ER streaming in growing root hairs is only $\sim 0.3 \mu\text{m}/\text{sec}$ (Figure 7d, S3b). Furthermore, it is known that root hairs grow at a rate of approximately 0.015 - $0.02 \mu\text{m}/\text{sec}$ (Grierson et al., 2014), it would be difficult to count the fast actin-myosin dependent ER extension as a major driven force for ER dynamics in growing root hairs. Likely, different from epidermal leaf and hypocotyl cells, in growing root hairs, MT-dependent ER extension play a major role in ER dynamics. This view is also supported by our results that *rhd3*

genetically interacts with MTs but not actin (Qi et al., 2016). We have added this in the revised manuscript.

16. In Arabidopsis, ER tubules do not extend with growing microtubule ends but rather elongate along preexisting microtubules toward both plus and minus ends (Hamada et al. 2014). ARK1 localizes in growing plus ends of microtubules, so the result in Fig 7 indicates a new mode: ER elongation with growing plus ends. Please explain the discrepancy and provide quantification of the ratio of ARK1-positive ER extension events (growing-end dependent) versus ARK1-negative ER extension events (preexisting-microtubule-lattice dependent). This quantification will provide the contribution of ARK1-dependent ER extension mechanism in the whole ER network and elongation. The frequency of ER fusion events per encounter with other ER (in both ARK1-positive/negative ER ends) will be helpful to clarify functional significance of ARK1-dependent mechanism.

Hamada et al. (2014) used GFP-tubulin to label MTs. It is known that GFP-tubulin is not ideal to label plus end of MTs, so it is possible that in their study, plus end dependent ER extension events were missed. Furthermore, Hamada et al. (2014) used Lat-B to treat cells, such treatment may also alter the ER extension.

When ARK1-GFP is expressed in leaf epidermal cells, it also often marks MTs, so we quantified the ratio of ARK1-positive, growing-end dependent ER extension events versus preexisting-microtubule-lattice dependent ER extension events. Of 116 events examined, there are 28 ARK1-positive, growing-end dependent ER extension events and 88 preexisting-microtubule-lattice dependent ER extension events. Our data suggested that, in leaf epidermal cells, ARK1-positive, MT growing-end dependent ER extension contributes to a certain portion of MT-dependent ER elongation. We have added this in the revised manuscript.

17. Cortical microtubule lattice provides the anchoring sites of ER tubules, which act as the branching sites of ER (Hamada et al. 2014). The plus-end ARK1-dependent extension promotes encounter of ER ends with other ER structure, and RHD3 physically interacted with ARK1 induces ER fusion. The former mechanism increases ER network complexity and ER free ends, whereas the latter reduces complexity and free ends. I wonder functional significance of ARK1-dependent extension mechanism in root hair growth (how this mechanism contributes ER organization, especially in subapical ER network of root hairs). And why is ARK1 used for ER extension-fusion mechanism?

As we pointed in our response to the comment #15, we believe that, MT-dependent extension plays a major role in the growth of root hairs, as fast actin-myosin dependent ER extension would be too fast for the ER streaming ($\sim 0.3 \mu\text{m}/\text{sec}$ (Figure 7d, S3b)) and the growth rate of root hairs at $0.015\text{-}0.02 \mu\text{m}/\text{sec}$ (Grierson et al., 2014). As the distribution and dynamics of RHD3 is altered in growing root hairs of *ren9-1*, we think that ARK1-dependent tip growth of the ER is important for the subapical focused ER distribution and dynamics in growing root hairs. We agree that, cortical microtubule lattice in the subapical region would provide anchoring sites for

ER tubules, which act as the branching sites of the ER. ARK1 would then lead the movement of the ER in the cortex of subapical region, together with RHD3. This cortical fine ER network formed would then provide a basis for the extension of the ER into the endoplasm. Without ARK1, the movement of RHD3 is uncontrolled and mis-localized, so RHD3 is seen in the tip region as well as in the region behind the subapical region (Figure 7a-c). Similarly, cortical RHD3 that overlaps with ARK1 may also hold ARK1 to provide a basis for ring-like ARK1 formation in the subapical region. As in the *rhd3* mutant background, ARK1 tends to move to the tip region. It is known that in root hair growth, both MTs and ER are required for correct growth direction (Qi et al., 2016), the ARK1-RHD3 complex may serve as a bridge between MTs and ER. We have added this view into the revised manuscript.

18. Microtubule-stabilizing compound, taxol, promotes *ark1* phenotype, whereas microtubule-depolymerizing compound, oryzalin, partially rescues *ark1* (Eng and Wasteneys 2014). This supports that ARK1 increases microtubule catastrophe and free tubulin concentration to enhance new microtubule elongation. However, both taxol and oryzalin partially rescue *rhd3* phenotype (Qi et al. 2016). Please explain the discrepancy of inhibitor effects.

As we have pointed out in Qi et al., 2016, there is a concentration-dependent effect of taxol on *rhd3*. The rescue effect of taxol on *rhd3* phenotype is only observed when taxol is at 0.5 μ M or less, while in Eng and Wasteneys (2014) and this study, taxol concentration is 1 μ M or greater. In human neuron cells, taxol at low dosage (10 nM) also partially rescue atlastin (the RHD3 homolog) mutant neuron cells (Zhu et al., 2014). The dosage-dependent taxol effect was also observed in other mammalian cells (Derry et al., 1995).

Reviewer #3 (Remarks to the Author):

.....

Figure 4b: the co-localisation here is really hardly visible and the movie is very fussy. The mCherry-ARK fusion seems to show less comets/punctae than a GFP version?

It is true that ARK1-RFP marks less comets than the GFP version as ARK1-RFP is not as bright as ARK1-GFP. Because the data in the original Figure 4b and Figure 51-c were kind of redundant, we decided to only use ARK1-GFP in this revised manuscript. For better visualization, we did additional 3D analysis of ARK1-GFP and RFP-RHD3 (see our response to the next comment).

Figure 5 a,b: again here I struggle to see how ARK1 is surrounding RHD3 due to a lack of resolution.

To better illustrate the spatial relationship between ARK1 and RHD3, as suggested by the reviewer #1, we have constructed a 3D movie from a z-stack in a growing root hair. As shown in

Figure 6d and supplemental movies 7-9, while ARK1-GFP is enriched in the cortical region and localized in the plus end of microtubules (Figure 5c, supplemental Movies 2-4), RFP-RHD3 is distributed in the endoplasm as well as in the cortical region. Clearly, ARK1-GFP partially overlaps with subapical focused RFP-RHD3 in the cortical region, which confirmed what we reported in Figure 6b and supplemental Movie 6.

Figure 5d: I cannot see a real co-localisation of the ARK1 bundles with RHD3, rather the opposite in at least two cases where a bundle of ARK1 has no RHD3 label.

The partial association of bundled ARK1-GFP with RFP-RHD3 in the endoplasm of taxol (1 μ M) treated cells we talked is relative to ARK1-GFP and RFP-RHD3 in non-treated root hairs. After the Taxol treatment, we revealed there is an increased association of ARK1-GFP and RFP-RHD3 (yes, it is still partial) in the endoplasm, while in non-Taxol treated samples, the association is very hard to find, if it is still possible. We have included this in supplemental figure S3.

Figure 6a: the Western blot background should not be removed with the bands exaggerated. The size marker should be on the same gel or sizes can be indicated by labeling.

We have re-run the coIP and western blot with an added negative control ARK1 Δ ARM-GFP. The result is presented in Figure 8a. The sizes of marker are indicated by labeling.

Figure 6b: I cannot see any clear ER labeling with any of the combinations. This figure is insufficient for publication.

BiFC of ARK1-RHD3 does not label the ER but ER-microtubule junctions proximal to the ER, while ARK1 Δ ARM-RHD3 tends to be on the ER as punctates, as indicated by mCherry-HDEL. To improve the quality of images, we have repeated BiFC and have now replaced the old ones with a new set of data.

Figure 7: I appreciate that imaging in root hair cells has its difficulties but this is now done in tobacco leaves and therefore the imaging resolution can be improved. Especially (but not only) in a printed version the images are very unclear and pixelated.

Although the ARM deletion shows no interactions, the authors should comment why only 10 out of 30 co-move. I can also not find how many biological and technical replicas were used.

In general, the ER in epidermal cells of leaves can be imaged in high resolution. However, in this experiment, we need to collect frames at least 1 per 0.873 second to track the movement of ARK1 and RHD3. We found that it is difficult to collect high-resolution ER images while tracking the elongation of ER tubules. Also, we prepared our figures in large in the width of 8.5 inch at this stage, so images may look pixelated when printed. Nevertheless, to improve the resolution of the ER, we did a deconvolution of images collected.

The initiate quantification of the numbers of ARK1 that moves together with RHD3 at tips of ER tubules vs. don't (10/30) came from 5 technical replications, each time 5 cells were examined. In this revised manuscript, we examined in 4 additional technical replications, 5 cells in each replication. We found that 18 out of 50 ARK1 comets move together with RHD3 (total number now would be $28(10+18)/80(30+50)$), while still no ARK1 Δ ARM comets moves with RHD3. With regard to why only a portion of ARK1 comets move together with RHD3, we think that it may be a reflection of a weak activity of ARK1 in mature leaf epidermal cells. ARK1 is predominantly expressed hence mainly active in growing root hairs. It is also possible that, in leaf epidermal cells, other factors may compete with ARK1 to guide the action of RHD3.

The ARK1 and Δ ARM labelling looks rather different. Is that representative?

Yes, there is a degree of difference between the two situations. When transient expressed in tobacco leaf epidermal cells, ARK1 labels plus ends as well as MTs, while ARK1 Δ ARM is seen at plus ends, and as punctae on MTs and in the cytosol as well. Such labeling of ARK1 and ARK1 Δ ARM is representative. We repeated the experiments 6 times, and observed over 100 cells. We have added this into the revised manuscript. We tracked the movement of growing ends (plus ends) for both ARK1 and ARK1 Δ ARM to avoid the discrepancy.

Figure S1 is lacking labelling a-e

Also the dataset is not fully clear to me and would really need a better and more in-depth figure legend

More detailed figure legend has been added to explain the different oscillations of YFP-RAB-A2a in 4 different genotypes.

Figure S4 should be combined with Figure 3.

We have combined Figure S4 with Figure 3. In addition, we have added our quantification of ER and MT distributions as suggested by the reviewer #2.

Line 76: here Sparkes et al 2010 "Five Arabidopsis Reticulon Isoforms Share Endoplasmic Reticulum Location, Topology, and Membrane-Shaping Properties" should be cited for the RTN topology investigation

The citation has been added.

Line 239: the statement that an OD of 0.01 is not considered overexpression is not correct. I appreciate that overexpression is being a difficult terminology but given that there is additional expression of a non-native protein this statement is misleading and should be removed.

We have removed this statement.

Line 631 “of plants indicated”: please clarify the title

We meant to refer 4 different plant genotypes examined in Figure S4. We have revised the title accordingly.

References cited:

- Chen, J., Doyle, C., Qi, X., and Zheng, H. (2012). The endoplasmic reticulum: a social network in plant cells. *J Integr Plant Biol* *54*, 840-850.
- Derry, W.B., Wilson, L., and Jordan, M.A. (1995). Substoichiometric binding of taxol suppresses microtubule dynamics. *Biochemistry* *34*, 2203-2211.
- Eng, R.C., Halat, L.S., Livingston, S.J., Sakai, T., Motose, H., and Wasteneys, G.O. (2017). The ARM Domain of ARMADILLO-REPEAT KINESIN 1 is Not Required for Microtubule Catastrophe But Can Negatively Regulate NIMA-RELATED KINASE 6 in *Arabidopsis thaliana*. *Plant Cell Physiol* *58*, 1350-1363.
- Eng, R.C., and Wasteneys, G.O. (2014). The microtubule plus-end tracking protein ARMADILLO-REPEAT KINESIN1 promotes microtubule catastrophe in *Arabidopsis*. *Plant Cell* *26*, 3372-3386.
- Galway, M.E., Heckman, J.W., Jr., and Schiefelbein, J.W. (1997). Growth and ultrastructure of *Arabidopsis* root hairs: the *rhd3* mutation alters vacuole enlargement and tip growth. *Planta* *201*, 209-218.
- Grierson, C., Nielsen, E., Ketelaarc, T., and Schiefelbein, J. (2014). Root hairs. *Arabidopsis Book* *12*, e0172.
- Hamada, T., Ueda, H., Kawase, T., and Hara-Nishimura, I. (2014). Microtubules contribute to tubule elongation and anchoring of endoplasmic reticulum, resulting in high network complexity in *Arabidopsis*. *Plant Physiol* *166*, 1869-1876.
- Qi, X., Sun, J., and Zheng, H. (2016). A GTPase-Dependent Fine ER Is Required for Localized Secretion in Polarized Growth of Root Hairs1. *Plant Physiol* *171*, 1996-2007.
- Robida, A.M., and Kerppola, T.K. (2009). Bimolecular fluorescence complementation analysis of inducible protein interactions: effects of factors affecting protein folding on fluorescent protein fragment association. *J Mol Biol* *394*, 391-409.
- Stefano, G., and Brandizzi, F. (2014). Unique and conserved features of the plant ER-shaping GTPase RHD3. *Cell Logist* *4*, e28217.
- Van Bruaene, N., Joss, G., and Van Oostveldt, P. (2004). Reorganization and in vivo dynamics of microtubules during *Arabidopsis* root hair development. *Plant Physiology* *136*, 3905-3919.
- Zhu, P.P., Denton, K.R., Pierson, T.M., Li, X.J., and Blackstone, C. (2014). Pharmacologic rescue of axon growth defects in a human iPSC model of hereditary spastic paraplegia SPG3A. *Hum Mol Genet.*

REVIEWERS' COMMENTS

Reviewer #1 (Remarks to the Author):

The authors have adequately addressed the most of the important issues that I raised in the initial review of this manuscript. I still think it would be helpful for the authors to attempt to provide another kinesin motor protein control in figure 8a to demonstrate specificity of the RHD3 interaction, but this is likely a relatively minor issue. Other than that my main concern regarding the quality of the data presented to establish colocalization between RHD3 and ARK1 at the tips of root hair MTs have been addressed.

Reviewer #2 (Remarks to the Author):

Thank you for your revision and comments.
I agree with your comments and additional analyses.
I wish publication of this article in Nature Comm.

Reviewer #3 (Remarks to the Author):

Revision Review for: Sun et al. "ARK1, a kinesin protein acts with RHD3 to link microtubules to the ER for generation of a fine ER network in Arabidopsis"

The authors have addressed all my comments to their best possibilities.
I would ask though to double-check that answers to reviewer comments are also incorporated into the manuscript for clarification.
Also it would be worth reading over the manuscript again and check on grammar/spelling/full sentences.

The point-by-point response to remaining reviewer comments

REVIEWERS' COMMENTS

Reviewer #1 (Remarks to the Author):

The authors have adequately addressed the most of the important issues that I raised in the initial review of this manuscript. I still think it would be helpful for the authors to attempt to provide another kinesin motor protein control in figure 8a to demonstrate specificity of the RHD3 interaction, but this is likely a relatively minor issue. Other than that my main concern regarding the quality of the data presented to establish colocalization between RHD3 and ARK1 at the tips of root hair MTs have been addressed.

ARK1 Δ ARM is localized to the microtubule plus-end and it even rescues microtubule dynamics of *ark1* (Eng et al., 2017), so ARK1 Δ ARM is a specific negative control in this study. We believe that Reviewer 1 also appreciated this. We agree that two negative controls are better than one. However, it is time-consuming to find a kinesin candidate in this study. We will study this in our future study.

Reviewer #2 (Remarks to the Author):

Thank you for your revision and comments.
I agree with your comments and additional analyses.
I wish publication of this article in Nature Comm.

Thank you for these positive comments.

Reviewer #3 (Remarks to the Author):

Revision Review for: Sun et al. "ARK1, a kinesin protein acts with RHD3 to link microtubules to the ER for generation of a fine ER network in Arabidopsis"

The authors have addressed all my comments to their best possibilities.
I would ask though to double-check that answers to reviewer comments are also incorporated into the manuscript for clarification.
Also it would be worth reading over the manuscript again and check on grammar/spelling/full sentences.

Thanks. We checked again our revision in the text and also spelling and grammar to minimum errors, while we are revising the manuscript according to the editorial requests.